

# Quantification of peroxynitric acid and peroxyacyl nitrates using an ethane-based thermal dissociation peroxy radical chemical amplification cavity ring-down spectrometer

Youssef M. Taha[1], Matthew T. Saowapon[1], Faisal V. Assad[1], Connie Z. Ye[1], Xining Chen[1,a], Natasha M. Garner[1], and Hans D. Osthoff[1]

[1] Department of Chemistry, University of Calgary, 2500 University Drive N.W., Calgary, Alberta, Canada T2N 1N4
[a] now at: Department of Chemistry, McGill University, 801 Sherbrooke St. West, Montreal, Quebec, Canada H3A 2K6

*Correspondence to*: Hans D. Osthoff (hosthoff@ucalgary.ca)

**Abstract.** Peroxy and peroxyacyl nitrates (PNs and PANs) are important trace gas constituents of the troposphere which are challenging to quantify by differential thermal dissociation with $NO_2$ detection in polluted (i.e., high-$NO_x$) environments. In this paper, a thermal dissociation peroxy radical chemical amplification cavity ring-down spectrometer (TD-PERCA-CRDS) for sensitive and selective quantification of total peroxynitrates ($\Sigma PN = \Sigma RO_2NO_2$) and of total peroxyacyl nitrates ($\Sigma PAN = \Sigma RC(O)O_2NO_2$) is described. The instrument features multiple detection channels to monitor the $NO_2$ background and the $RO_x$ (= $HO_2 + RO_2 + \Sigma RO_2$) radicals generated by TD of $\Sigma PN$ and/or $\Sigma PAN$. Chemical amplification is achieved through addition of 0.6 ppm NO and 1.6% $C_2H_6$ to the inlet. The instrument's performance was evaluated using peroxynitric acid (PNA) and peroxyacetic or peroxypropionic nitric anhydride (PAN or PPN) as representative examples of $\Sigma PN$ and $\Sigma PAN$, respectively, whose abundances were verified by iodide chemical ionization mass spectrometry (CIMS). The amplification factor or chain length increases with temperature up to 69±5 and decreases with analyte concentration and relative humidity (RH). At inlet temperatures above 120 °C and 250 °C, respectively, PNA and $\Sigma PAN$ fully dissociated, though their TD profiles partially overlap. Furthermore, interference from ozone ($O_3$) was observed at temperatures above 150 °C, rationalized by its partial dissociation to O atoms which react with $C_2H_6$ to form $C_2H_5$ and OH radicals. Quantification of PNA and $\Sigma PAN$ in laboratory-generated mixtures containing $O_3$ was achieved by simultaneously monitoring the TD-PERCA responses in multiple parallel CRDS channels set to different temperatures in the 60 °C to 130 °C range. The (1 s, 1 σ) limit of detection (LOD) of TD-PERCA-CRDS is 3.4 pptv for PNA and 1.3 pptv for $\Sigma PAN$ and significantly lower than TD-CRDS without chemical amplification. The feasibility of TD-PERCA-CRDS for ambient air measurements is discussed.





## 1 Introduction

The $RO_x$ (= OH + $HO_2$ + $\Sigma RO_2$) radicals and the nitrogen oxides ($NO_x$ = NO + $NO_2$) are important trace constituents of the atmosphere that drive diverse processes such as the photochemical production of ozone ($O_3$) in the troposphere (Kirchner and Stockwell, 1996; Fleming et al., 2006), the catalytic destruction of $O_3$ in the stratosphere (Bates and Nicolet, 1950; Stenke and Grewe, 2005; Solomon, 1999; Portmann et al., 1999), and the chemistry of organic aerosol formation (Ziemann and Atkinson, 2012; Ehn et al., 2014; Crounse et al., 2013). In the troposphere,

the concentrations of these species are frequently buffered by $RO_x$ and $NO_x$ reservoir species, of which peroxynitric acid (PNA, $HO_2NO_2$), alkyl peroxy nitrates such as methyl peroxynitrate ($CH_3O_2NO_2$, MPN), and the peroxyacyl nitrates (PANs, $RC(O)O_2NO_2$) are important examples (Singh et al., 1992; Roberts, 1990). Much insight into $RO_x$ and $NO_x$ chemistry has been gained by measuring the atmospheric abundances of these reservoirs. Significant PNA concentrations, for example, have been observed in the polar regions (Slusher et al., 2001; Davis et al., 2004; Jones

et al., 2014), aloft in the free and upper troposphere (Kim et al., 2007; Murphy et al., 2004), in highly polluted environments such as the Uintah basin in winter (Veres et al., 2015), and in urban atmospheres (Spencer et al., 2009; Chen et al., 2017), and have given valuable insights into radical budgets at these locations. The PANs and PNA are prone to thermal dissociation, such that higher concentrations are more commonly observed in cold regions, aloft in the free / upper troposphere, or in winter (Table 1). On the other hand, their rate of production is

greater in summer as the actinic flux intensifies. Mixing ratios of PNA peak in the range from 10s to a few 100s of parts-per-trillion by volume (pptv, $10^{-12}$) and those of peroxyacetic nitric anhydride (PAN; $CH_3C(O)O_2NO_2$) can exceed ten parts-per-billion by volume (ppbv, $10^{-9}$) (Tuazon et al., 1981).

There is ongoing interest to develop improved techniques for quantification of PANs (Roberts, 2007; Wooldridge et al., 2010; Zheng et al., 2011; Tokarek et al., 2014) and PNA (Murphy et al., 2004; Spencer et al., 2009; Veres et

al., 2015; Chen et al., 2017). Thermal dissociation (TD) methods such as TD coupled to laser-induced fluorescence (TD-LIF) (Wooldridge et al., 2010; Day et al., 2002; Di Carlo et al., 2013), to cavity ring-down spectroscopy (TD-CRDS) (Paul et al., 2009; Thaler et al., 2011; Paul and Osthoff, 2010; Thieser et al., 2016; Sobanski et al., 2016; Womack et al., 2017), or to cavity phase-shift spectroscopy (TD-CAPS) (Sadanaga et al., 2016) detection of $NO_2$ are attractive as they can be used to monitor all components of odd nitrogen ($NO_y$) in parallel, including $NO_2$ (inlet

operated at ambient temperature), total peroxy nitrates ($\Sigma PN$ = PNA + MPN + ...; inlet heated to ~100 °C), total peroxyacyl nitrates ($\Sigma PAN$ = PAN + peroxypropinoic nitric anhydride ($C_2H_5C(O)O_2NO_2$ + ...; ~ 250 °C), total alkyl nitrates + $ClNO_2$ ($\Sigma AN$; ~ 420 °C), and $HNO_3$ (~ 600 °C), simply by deploying multiple detection channels



and setting appropriate inlet temperatures. Molecules such as NO and HONO can be quantified through addition of $O_3$ following TD (Wild et al., 2014; Womack et al., 2017; Fuchs et al., 2009).

When used in polluted (i.e., high-$NO_x$) environments, however, a drawback of the TD methods is that quantification of $\Sigma PN$ (and, to a lesser degree, also of $\Sigma PAN$) is compromised because of the large error introduced from subtraction of the $NO_2$ background, which is often $2 - 4$ orders of magnitudes larger than the $\Sigma PN$ abundance. In such high-$NO_x$ environments, on the other hand, mixing ratios of $RO_x$ radicals are generally much smaller, < 100 pptv (Wood et al., 2016), than those of $NO_2$, such that a better strategy may be to quantify the peroxy and
peroxyacyl radicals generated in stoichiometric amounts during TD of $\Sigma PN$ and/or $\Sigma PAN$ rather than $NO_2$.

The $RO_x$ radicals may be quantified by chemical ionization mass spectrometry (CIMS) (Hanke et al., 2002; Edwards et al., 2003; Chen et al., 2004; Slusher et al., 2004; Hornbrook et al., 2011); in fact, a TD-CIMS method has been developed to quantify PAN by titrating the peroxyacetyl radical with iodide reagent ion (Slusher et al., 2004). Other $RO_x$ radical detection methods include LIF (Faloona et al., 2004; Heard, 2006; Fuchs et al., 2008;
Dusanter et al., 2009) and peroxy radical chemical amplification (PERCA) coupled to $NO_2$ detection (Cantrell et al., 1984; Hastie et al., 1991; Green et al., 2006; Liu and Zhang, 2014; Horstjann et al., 2014). The PERCA method is attractive as it allows the infrastructure of existing TD instruments with $NO_2$ detection to be utilized.

In PERCA coupled to $NO_2$ detection, concentrations of $RO_x$ radicals are amplified by factors of between ~20 to ~190 through a series of catalytic reactions, usually involving parts-per-million by volume (ppmv, $10^{-6}$) mixing
ratios of nitric oxide (NO) and percent levels of either carbon monoxide (CO) (Cantrell et al., 1984) or a short chain hydrocarbon such as ethane ($C_2H_6$) (Mihele and Hastie, 2000; Wood et al., 2016) (R1-R6, Table 2). Under these conditions, the peroxy radicals catalytically convert NO to $NO_2$, and the amount of $NO_2$ produced over a constant reaction period is proportional to the number of radicals that were present originally.

The measurement of peroxy radicals by PERCA is prone to interferences. For instance, a key operational parameter
of any PERCA instrument is the radical chain length or amplification factor (CL), which must be carefully calibrated. This chain length is suppressed by water vapour, whose presence increases the rates of radical loss on the inner walls of the PERCA chamber and the rates of certain gas-phase reactions, e.g., the reaction between the hydroperoxyl radical - water dimer ($HO_2 \cdot H_2O$) with NO to peroxynitrous acid (HOONO) which isomerizes to nitric acid ($HONO_2$) (Mihele and Hastie, 1998; Mihele et al., 1999 ; Mihele and Hastie, 2000).

The most obvious way to minimize wall reactions and to prevent weakly bound clusters such as $HO_2 \cdot H_2O$ from forming is to apply heat. When quantification of ambient $RO_x$ radicals is the goal, this is avoided to prevent TD of $\Sigma PN$ or $\Sigma PAN$ (which are more abundant than free $RO_x$ radicals) as these molecules would interfere (Mihele and Hastie, 2000). On the other hand, if measurement of $\Sigma PN$ or $\Sigma PAN$ is desired (such as in this paper), this



interference is turned into a measurement principle. We are aware of only one prior attempt to quantify peroxy
nitrates in this manner: Blanchard et al. thermally decomposed PAN eluting from a chromatographic column in the
presence of NO and CO and quantified the amplified $NO_2$ using luminol chemiluminescence (Blanchard et al.,
1993).

In this manuscript, we probe the feasibility of selectively quantifying $\Sigma PN$ and $\Sigma PAN$ through their respective
peroxy radical TD fragments by thermal decomposition peroxy radical chemical amplification cavity ring-down
spectroscopy (TD-PERCA-CRDS). The instrument uses a 405 nm blue diode laser CRDS (Paul and Osthoff, 2010)
to monitor $NO_2$ and the ethane-based chemical amplification scheme described by Wood et al. (2016), chosen
because of ethane's lower toxicity compared to CO. Thermal dissociation profiles were determined for PAN, PPN,
and PNA. Quantification of trace levels of PAN, PPN, and PNA by TD-PERCA-CRDS is demonstrated and
compared to parallel measurements by iodide CIMS. The suitability of TD-PERCA-CRDS as a highly sensitive
(sub-pptv) $\Sigma PN$ and $\Sigma PAN$ detection method for ambient measurements is discussed.

## 2 Experimental Section

### 2.1 TD-PERCA-CRDS

#### 2.1.1 Single-channel inlet

The majority of the experiments described in this manuscript were conducted using a single-channel TD-PERCA
inlet that has been described elsewhere (Taha et al., 2018). Briefly, NO in $N_2$ (100.2 ppmv, Scott-Marrin, Riverside,
CA) was scrubbed of $NO_2$ by passing through iron(II) sulfate heptahydrate (99%, Sigma-Aldrich, Oakville, ON)
prior to being combined with either a flow of $N_2$ gas (Praxair) or ethane (CP grade, 99%, Matheson, Baskin Ridge,
NJ). The gas mixture was directed towards the 80-cm long PERCA chamber (1.27 cm or ½" outer diameter (o.d.)).
When NO and $N_2$ were mixed prior to the PERCA chamber, a "PERCA off" signal was observed, and ethane was
added after the PERCA chamber to maintain constant flow (and pressure) through the system. Conversely, when
NO and ethane were directed towards the PERCA chamber a "PERCA on" signal was observed. To maintain flows
and pressures through the system during "PERCA on" mode, $N_2$ was added to the inlet at an addition point after
the PERCA chamber.

The inlet was connected to a four-channel CRDS described elsewhere (Odame-Ankrah, 2015). Briefly,
concentrations of $NO_2$ were monitored via its absorption at 405 nm (Paul and Osthoff, 2010). A flow containing
ppmv levels of $O_3$ in $O_2$ was added to one CRDS channel to monitor $NO_x$ (Fuchs et al., 2009). Each channel was
connected to a MFC set to a flow rate of ~0.84 slpm. All four CRDS cells were connected to sample the gases



exiting the single-channel TD-PERCA inlet, though in principle a single CRDS detection channel would have suffied to carry out the measurements.

When radical free "zero" air was sampled with the single channel inlet, a negative offset (up to 1 ppbv) was observed when the 3-way solenoid valves were switched between PERCA "on" and "off" (data not shown). This artefact was not observed with the dual channel setup (section 2.1.2) and was not further investigated but may have been caused by differences in the ethane flows through the needle valve induced by pressure changes (up to 15 Torr) during switching between PERCA on and off.

**2.1.2 Dual-channel TD-PERCA-CRDS**

Time resolution, signal-to-noise, and subtraction of background $NO_2$ can be significantly improved in a PERCA instrument by implementing dual detection channels where both amplified and background signals are simultaneously monitored (Green et al., 2006; Cantrell et al., 1996). The dual channel TD-PERCA setup used in this work is shown in Figure 1. The NO reagent gas is added at the same flow rate and concentration as in the

single-channel version near the tip of the inlet (after the zero air and calibration gas ports). The residence time prior to the ethane addition point (~2.0 s) suffices to destroy $RO_x$ radicals (via reaction with NO) prior to chemical amplification. Addition of either NO or ethane is on/off modulated using 2-way normally-open valves connected to a pump via 50 μm critical orifices (Lenox Laser, Glen Arm, MD) in a similar fashion as described earlier (Odame-Ankrah and Osthoff, 2011).

A portion of the sample flow of ~0.74 slpm was diverted prior to TD to monitor the "background" $NO_2$ concentration in one CRDS channel. The remaining flow (~2.2 slpm) was passed through the heated quartz tube and PERCA chamber as described earlier and sampled by the other three CRDS channels to monitor background $NO_2$ plus the amplified $NO_2$ signal.

**2.1.3 Four-channel differential temperature TD-PERCA-CRDS**

An instrument with four identical measurement channels was set up to enable simultaneous quantification of $NO_2$, $NO_2 + \Sigma PN$ and $NO_2 + \Sigma PN + \Sigma PAN$. This version was similar to dual channel setup described above and simply added two additional PERCA chambers; all three PERCA chambers were connected to separate CRDS channels. The quartz PERCA chamber was replaced with three identical 1.27 cm (1/2") o.d. and 0.95 cm (3/8") i.d. Teflon™ tubes externally heated using stretch-to-fit heaters (Watlow, St. Louis, MO) to 60, 80 and 100 °C, respectively. A

common inlet filter was placed between the PERCA chambers and the ethane addition point. A flow restriction




was placed ahead of the common filter to achieve a pressure of 380 - 400 Torr at a flow rate of 0.8 slpm per channel (total flow rate ~3.2 slpm).

## 2.2 Synthesis and delivery of PAN and PPN

The synthesis of PAN and PPN from their corresponding anhydrides was described earlier (Mielke and Osthoff, 150    2012; Furgeson et al., 2011). Aliquots in tridecane were stored in 2.0 mL centrifuge tubes (VWR) in a freezer until needed.

To separate PAN and PPN from impurities generated during synthesis and storage (i.e., $NO_2$, $HNO_3$ and alkyl nitrates (Grosjean et al., 1994)), a preparatory scale GC setup (Figure 2) was used. First, the contents of two tubes containing PAN and PPN in tridecane were combined in a 3-valve glass vessel. This vessel was connected to a 2-155    position GC-Valve (VICI Valco EH4C10WE, Houston, TX) and mildly pressurized (~0.1 atm above ambient) using oxygen (Praxair) delivered through a 10 μm critical orifice. Two megabore capillary gas chromatography (GC) columns (Restek RTX-1701, 0.53 mm i.d., 1.00 μm film thickness, State College, PA) of equal lengths (3 m) were connected to the ports adjacent to the one connected to the glass vessel. The outlet of one GC column was connected to the PERCA inlet, whereas the other was exhausted into a waste line. The port opposite to the glass 160    vessel was also pressurized with oxygen such that both columns remained under flow at all times. Gases were delivered by switching the valve to position A (Figure 2) for ~40 s. The output of the preparatory-scale GC was diluted with ZA or air passed through a custom-built scrubber system to meet the sample flow requirements of the TD-PERCA CRDS and/or CIMS. The relative humidity (RH) of the gases delivered was monitored using a temperature/RH probe (VWR) placed inline.

The preparatory scale GC setup allowed delivery of short "bursts" of PAN and PPN. To deliver a constant and low PAN concentration over prolonged time periods, air was drawn from a 4,000 L Teflon chamber, initially filled with scrubbed (i.e., PAN-free) air and to which the head space above a PAN/tridecane solution had been added. An internal mixing fan ensured constant output.

## 2.3 Synthesis and delivery of PNA

**2.3.1. Batch sample**

A batch PNA sample was synthesized from reaction of nitronium tetrafluoroborate ($NO_2BF_4$; Sigma-Aldrich) with $H_2O_2$ as described by Chen et al. (2017). Briefly, a 50% $H_2O_2$ solution (Sigma-Aldrich) was concentrated using a gentle $N_2$ flow over a period of several days. A small aliquot (200 μL) of concentrated $H_2O_2$ was placed in a 3-valve glass vessel cooled to 0 °C using an external circulating chiller, and 120 mg of $NO_2BF_4$ were added. The





headspace of the glass vessel was flushed with a 50 sccm flow of $N_2$ delivered by a MFC. This flow delivered very high concentrations and contained substantial and variable amounts of impurities (mainly $HNO_3$), even when the vessel temperature was lowered to -20 °C. The batch sample was used to calibrate the CIMS against TD-CRDS (section 2.4).

### 2.3.2. Photolysis source

Gas flows containing low and reproducible concentrations of PNA were generated dynamically in a similar fashion to the method described by Veres et al. (2015) by combining the output of a $HO_2$ photochemical source with $NO_2$. Ultrapure $N_2$ (Praxair) was passed through a bubbler filled with deionized water at a flow rate of 100 sccm and combined with 2 sccm of $O_2$ (Praxair). This mixture was passed through a ¼" (0.635 cm) o.d. quartz tube partially illuminated by a low-pressure 185 nm mercury quartz lamp (Jelight 95-2100-2, Irvine, CA). This generated a mixture of $O_3$ (~30 ppbv after dilution), OH and $HO_2$, whose concentrations were controlled with a sliding metal sleeve (VWR) which modified the length of the quartz tubing that was illuminated. This flow was combined with between 4 and 16 ppmv $NO_2$ to yield a gas mixture containing PNA which was immediately (< 5 cm tube length) diluted with zero or scrubbed air flowing at a rate slightly greater than the amount sampled by the instruments. The $NO_2$ gas stream was generated by mixing between 1.4 and 3.0 sccm of NO (100.2 ppmv in $N_2$; Scott-Marrin, Riverside, CA) with a slightly less than stoichiometric amount of $O_3$ in ~20 sccm $O_2$, generated by illuminating $O_2$ with a low-pressure 254 nm mercury quartz lamp (Jelight).

### 2.4 Chemical ionization mass spectrometry

The CIMS and its operation have been described elsewhere (Mielke et al., 2011; Mielke and Osthoff, 2012; Abida et al., 2011). For measurements of PAN or PPN, the instrument was operated with iodide reagent in declustering mode (collisional dissociation chamber voltage = -24.7 V) and sampled through a short section of 1.27 cm (½") o.d. PFA Teflon™ tubing heated to 190 °C. The inlet flow was diluted with nitrogen saturated with water vapour to maintain a minimum RH of ~16% with the ion-molecule reaction region (IMR). PAN and PPN were quantified using the acetate and propionate ions ($m/z$ 59 and 73). Ion counts were normalized to $10^6$ reagent ion counts prior to presentation. The instrument response factor for PAN was calibrated against TD-CRDS (Mielke and Osthoff, 2012) and was 11±3 Hz pptv$^{-1}$.

For PNA measurements, the CIMS was operated with an ambient temperature inlet and in clustering mode (collisional dissociation chamber voltage = -8.9 V); under these conditions, ~$5\times10^5$ I$^-$ and ~$2\times10^4$ I·$H_2O$ ions were observed. Mixing ratios of PNA were monitored primarily using $NO_3^-$ at $m/z$ 62, which is formed via PNA




decomposition within the IMR (Abida et al., 2011). The $HNO_3 \cdot I^-$ and $HO_2NO_2 \cdot I^-$ clusters at $m/z$ 190 and $m/z$ 206
(Veres et al., 2015; Chen et al., 2017) were also monitored.

The PNA response factors were determined using TD-CRDS (i.e., without added PERCA gases) with its inlet
operated at 120 °C (Figure 3). Assuming that one equivalent of $NO_2$ is generated for each PNA molecule thermally
dissociated, the CIMS response factors, normalized to $10^6$ $I^-$ counts, were 34.7±0.2 Hz pptv$^{-1}$ and 0.023±0.002 Hz
pptv$^{-1}$ at $m/z$ 62 and 206, respectively. These response factors are consistent with calibration factors by other groups
(Veres et al., 2015; Chen et al., 2017), with the low response at $m/z$ 206 rationalized by the low number of $I^- \cdot H_2O$
ions. Even though the CIMS response at $m/z$ 62 is not specific (Abida et al., 2011), it was used in the laboratory
experiments presented here to monitor PNA rather than $m/z$ 206 because of its larger response factor and thus
higher sensitivity.

## 2.5 Box model simulations

Box model simulations were carried out using a subset of the MCM V3.3.1 obtained from
http://mcm.leeds.ac.uk/MCM (Jenkin et al., 1997; Saunders et al., 2003; Jenkin et al., 2012) and the Kinetic
Preprocessor (KPP) (Sandu and Sander, 2006) to aid in the interpretation of observations. Details are given in the
S.I.

## 3 Results

### 3.1 Thermal dissociation profiles

The TD profiles of PNA, PAN, and PPN were measured by TD-CRDS (i.e., without amplification) with the single-
channel inlet and are shown in Figure 4. The superimposed trend lines are simulations based on the TD model
introduced by Paul et al. (2009) and the Arrhenius parameters in Table 3 and are consistent with the observations.
The TD profiles of PNA and PAN/PPN partially overlap and are consistent with the 5%/95% ranges given in Figure
3 of Wooldridge et al. (2010). PNA and PAN or PPN fully dissociated at temperatures of 120 °C and 250 °C,
respectively. These temperatures were used in subsequent experiments when complete dissociation of either PNA
or PAN/PPN was desired.

Also shown in Figure 4 is the TD-PERCA-CRDS signal observed when sampling $O_3$, an interfering species (see
section 3.6.1).



## 3.2 Measurement of PAN and PPN by TD-PERCA-CRDS

A time series demonstrating amplification of PAN and PPN in the TD-PERCA-CRDS operated with its inlet at 250 °C is shown in Figure 5. In this experiment, PAN and PPN were delivered via the preparatory-scale GC (Figure 2), and the single-channel setup (section 2.1.1) was used.

PAN and PPN eluted from the GC column after 3 min and 6 min, respectively. The compounds eluted as plateaus because of the relatively long injection time. In Figure 5A, PAN and PPN are observed only by the heated ($NO_2$+ $\Sigma$PAN) TD-CRDS channel, where they quantitatively (see Figure 5 of (Paul et al., 2009)) decompose to $NO_2$ at mixing ratios of 2.00±0.09 ppbv and 1.86±0.12 ppbv, respectively (errors are 1 $\sigma$ of 1 s data). After the PERCA heater was set to 250 °C as well, similar amounts of $NO_2$, 2.04±0.09 ppbv and 1.97±0.12 ppbv, were observed in the ambient temperature channel for PAN and PPN, respectively. Marginally higher amounts were observed in the heated CRDS channel (2.42±0.10 and 2.06±0.14 ppbv) (Figure 5B). The lower amounts observed in the unheated CRDS channel result from recombination of peroxyacyl radicals with $NO_2$ (mostly in the unheated PERCA chamber), which suppresses the signal in the unheated CRDS channel but not in the heated one. Hence, the $NO_2$ + $\Sigma$PAN data are a more accurate measure of the PAN and PPN concentrations delivered.

Because PAN and PPN dissociate with 1:1 stoichiometry, the amount of peroxyacyl radicals produced during thermal dissociation is the same as the amount of $NO_2$ generated. When ~0.75 ppmv of NO was added (Figure 5C), the peroxyacyl and, subsequently, the methyl (or ethyl) peroxy, and the hydroperoxyl radicals oxidize NO to $NO_2$ (reactions R8/R10, R1/R5, and R3; Table 2) and the $NO_2$ signal relative to the signal obtained in the absence of NO is amplified by a factor of four. The ratios observed (Figure 5C relative to Figure 5B) were 4.0±0.2 and 3.8±0.3 for PAN and PPN, respectively, and are consistent with earlier observations at lower NO mixing ratios (i.e., Figure 6 of Paul and Osthoff (2010)).

Next, NO and ethane were added at mixing ratios (0.75 ppmv NO and 1.5% $C_2H_6$) that Wood et al. (2016) determined to be optimal for ambient temperature PERCA. Under these conditions, the signals amplified to 116.0±1.3 ppbv and 109.3±0.7 ppbv (Figure 5D), corresponding to CLs (relative to Figure 4B) of 48±2 and 53±4 for PAN and PPN, respectively.

In the presence of ethane, marginally lower $NO_2$ concentrations (98.7% and 98.1%) were observed in the heated, $NO_2$ + $\Sigma$PAN TD-CRDS channel (compared to the absence of ethane). Partial scrubbing of $NO_2$ in heated quartz cells has been anecdotally observed in our group's and also others' (Womack et al., 2016) TD instruments; this effect varies between quartz cells and with sample history. Since the effect was relatively minor, it was neglected in this work.



### 3.3 Optimization of TD-PERCA amplification factors

Sequences, such as the one shown in Figure 5, were used to determine conditions leading to optimum amplification factors. The largest amplification factors were obtained with an ethane mixing ratio of ~1.6% - 1.7% (data not shown).

Figure 6 shows how the chain length varies with NO mixing ratio. In the absence of ethane, amplification factors of ~4 were observed (Figure 6, open symbols), consistent with the results shown in Figure 5C. When 1.7% ethane were added, the amplification factor increased with NO mixing ratio up to a maximum at 550±150 ppbv and then decreased, qualitatively consistent with the results reported by Wood et al. (2016).

The amplification factors shown in Figure 6 were slightly larger for PAN than for PPN mainly because the PPN mixing ratio of ~1.3 ppbv exceeded the optimum concentration range for PERCA (see section 3.5.2).

### 3.4 Parallel measurement of PNA by TD-PERCA-CRDS and CIMS

A sample time series showing TD-PERCA-CRDS measurements of photochemically generated PNA in scrubbed air is presented in Figure 7. Here, the TD-PERCA-CRDS was operated with the dual channel inlet (Figure 1) at 120 °C and with 1.6% $C_2H_6$ and 316±3 ppbv of NO (suboptimal NO mixing ratios). Figure 7A shows the $NO_2$ mixing ratios in the reference, $NO_2$ channel (grey trace) and in the TD-PERCA-CRDS channel (green trace). In this example, the mixing ratio of PNA was changed approximately every 2 min by moving a sliding cover within the photochemical source.

The difference between these two signals is displayed in Figure 7B (red trace, left-hand axis). Superimposed in Figure 7B (right-hand axis) are the CIMS responses at $m/z$ 62 ($NO_3^-$), $m/z$ 206 ($HNO_4 \cdot I^-$, multiplied by a factor of 100 for clarity), and $m/z$ 190 ($HNO_3 \cdot I^-$).

The photochemical source co-generates OH which is lost on the inner walls of the quartz tubing or is titrated by $NO_2$ or (to lesser extent since less abundant) NO to $HNO_3$ or HONO, respectively. Conceivably, the co-generation of $HNO_3$ could interfere with quantification of PNA by CIMS at $m/z$ 62. However, when the photolysis source was turned off at ~19:59 (Figure 7), $HNO_3$ was still observed for some time after at $m/z$ 190 due to slow desorption from the inner walls of the connecting tubing, whereas the ion counts at $m/z$ 62 quickly (< 10 s) returned to background values close to zero Hz, indicating that the contribution of $HNO_3$ to ion counts at $m/z$ 62 was negligible. The scatter plot of the TD-PERCA-CRDS and CIMS data at $m/z$ 62, multiplied by the CIMS response factor determined in Figure 3A, is shown in Figure 8A as dark blue circles. The signals by the two instruments are highly correlated ($r^2 = 0.979$), consistent with both instruments measuring the same molecule, PNA. The slope of this plot





(26.3±0.4) equals to the TD-PERCA-CRDS amplification factor for PNA. In contrast, the scatter plot of TD-
PERCA-CRDS with the CIMS response at *m/z* 206 was unusable because of the latter's poor signal to noise ratio
(Figure 3B).

## 3.5 Factors affecting amplification factors

The amplification factor for PNA shown in Figure 8A is less than observed for PAN in Figure 8B and for PAN and
PPN under optimal conditions (Figure 6). Though in this particular example the lower amplification factor was due
to the less than optimal amount of NO added, lower amplification factors for PNA than for PAN were generally
observed, even when optimum NO mixing ratios were used.

The obvious difference is that different radicals, $HO_2$ in the case of PNA and a peroxyacyl radical ($RC(O)O_2$) in
the case of PAN or PPN, are generated initially. However, in both cases, the $HO_2$/HO radical pair is the main carrier
of the amplification, such that this initial difference should only have a marginal effect. Wood et al. (2016)
estimated the uncertainty arising from the range in peroxy radical reactivity to ±9%.

Experimental parameters that can affect the amplification factor include relative humidity, radical concentration,
and the PERCA inlet temperature; these factors are probed separately in the following sections. In each case, box
model simulations were carried out (see S.I.) to aid in the interpretation of the data.

### 3.5.1 Dependence of chain lengths on relative humidity

We repeated the experiment described in section 3.4 with the scrubbed air humidified to 75% RH by passing the
make-up air through a bubbler. The resulting scatter plot is superimposed in Figure 8A as red squares. Indeed, the
amplification factor for PNA was lowered from 26.3±0.4 to 18.0±0.2 when the RH was increased from 20% to
75%. Following these observations, the RH dependence was investigated systematically with TD-PERCA-CRDS
operated under optimal conditions and with PAN and PPN at 250 °C inlet temperature. The results are summarized
in Figure 9.

The amplification factor decreased by (2.0±0.6)% for every 10% increase in RH. This RH dependence is less than
reported for ambient temperature PERCA: Between a RH of 0% and 50%, for example, the response of room
temperature PERCA dropped by 30% (Wood et al., 2016), whereas that of TD-PERCA decreased by 15%. A
reduced RH dependence is expected as the elevated temperature suppresses formation of $HO_2 \cdot H_2O$ (Kanno et al.,
2006), whose reaction with NO is a major radical sink (Mihele and Hastie, 1998; Mihele et al., 1999 ; Mihele and
Hastie, 2000). This interpretation is supported by box model simulations, which show a reduced RH dependence





of the CL at higher temperatures (Figure S-4). In addition, we speculate that reactions of radicals on the inner walls of PERCA tubing are reduced at higher inlet temperature.

**3.5.2 Dependence of chain lengths on radical concentration: dynamic range**

It is well known in the PERCA community that the chain lengths decrease at high radical concentrations due to radical-radical reactions. Figures 8A and 8B demonstrate that the response of TD-PERCA-CRDS is linear at low, atmospherically relevant mixing ratios (i.e., below ~0.6 ppbv). Figure 10 summarizes the PERCA responses as functions at larger PAN/PPN mixing ratios. The largest amplification factor, 69±5, was observed when the TD-PERCA inlet was operated at 250 °C with PAN or PPN mixing ratios ≤ 0.6 ppbv. Shorter chain lengths were

observed at higher mixing ratios (e.g., 62±2 at 1.3 ppbv, 53±4 at 2.1 ppbv, and 48±2 at 2.4 ppbv, respectively). Thus, the amplification factor is concentration dependent at $RO_2$ mixing ratios above ~0.6 ppbv and is constant under atmospherically relevant trace conditions.

PERCA reactors utilizing CO as a chain carrier show non-linearity at $RO_2$ mixing ratios above ~200 pptv (Hastie et al., 1991), while room temperature ethane based PERCA have a reported linear dynamic range up to ~800 pptv

(Wood et al., 2016). The greater dynamic range with ethane arises because of lower chain lengths and radical concentrations in the reactor and hence reduced radical-radical termination reactions (Wood et al., 2016). The linear range of the ethane TD-PERCA reactor of ~600 pptv falls in between these two extremes, as the CL and radical concentrations are greater than ethane PERCA at room temperature but less than those achievable with CO PERCA. The linear range observed is consistent with box model simulations, in particular when wall loss reactions are taken

into account (Figure S-6).

**3.5.3 Dependence of chain lengths on inlet temperature**

Next, we investigated the temperature dependence of the TD-PERCA-CRDS signal when sampling photochemically generated PNA at constant RH and PNA mixing ratio. Figure 11 shows such a temperature scan of ~180 pptv PNA (measured in parallel by CIMS). The non-amplified TD profile observed by TD-CRDS is

superimposed for comparison.

A striking feature in Figure 11 is the very large increase in the amplified $NO_2$ signal observed at temperatures above ~150 °C. This is an artefact that arises from $O_3$ co-emitted by the photochemical source and is commented on further in section 3.6.1.

It is obvious from Figure 11 that the amplification factor is strongly dependent on temperature: Even though PNA

fully dissociates at temperatures > ~90 °C in our inlets (Figure 4), the amplified signal increases by ~60% in the





region from 90 °C to 135 °C (Figure 11, insert), corresponding to amplification factors of ~15 and ~22, respectively. This increase is qualitatively consistent (if extrapolated) with the higher amplification factor observed with PAN or PPN at 250 °C.

Box model simulations using only gas-phase chemistry from the MCM V3.3.1 (Figure S-3) show that the CL is
expected to decrease with increasing temperature, opposite to what is observed. This occurs in the model because the chain-carrying reactions of $HO_2$ and $RO_2$ with NO (e.g., R3, Table 2) have negative activation energies and are hence slower at higher temperatures, yielding a lower CL at higher temperature. This is partially offset when the chemistry of $HO_2 \cdot H_2O$ is added to the mechanism (Figures S-4 and S-5) but does not suffice to achieve a higher CL at higher temperature; the latter is only predicted by the box model simulations if much lower wall loss reactivity
of OH and $HO_2$ are assumed (see S1.3 and S1.4).

### 3.6 Interferences

#### 3.6.1 Interference from $O_3$ in the measurement of ΣPAN at 250 °C

When sampling ambient air (data not shown) or when sampling photochemically generated PNA (Figure 11) the amplified $NO_2$ increases sharply at PERCA inlet temperatures above ~150 °C. These observations can be
rationalized by thermal decomposition of $O_3$. Even though only a small fraction of $O_3$ dissociates to $O_2 + O$ at ~150 °C in the TD inlet (~0.1%; Figure 5 (Jones and Davidson, 1962; Heimerl and Coffee, 1979)), a comparatively large signal is generated because the O atom reacts with $C_2H_6$ to form two radicals, OH and $C_2H_5$ (Baulch et al., 1994). This reaction is competitive in the PERCA inlet (compared to reaction of O with $O_2$) because of the high $C_2H_6$ concentration (1.7%): The lifetime of O with respect to reaction with $C_2H_6$ is ~0.34 ms, which is of similar
magnitude as the expected lifetime of O with respect to reaction with $O_2$ of ~0.15 ms (Hippler et al., 1990).

We considered an alternate inlet configuration in which the inlet length between the NO and ethane addition points is increased to allow for sufficient residence time completely titrate $O_3$ with the added NO. However, at the optimum NO mixing ratio for PERCA, the 1/e lifetime of $O_3$ is ~6 s, making this approach unfeasible.

Hence, if $O_3$ is sampled with an ethane-based TD-PERCA instrument heated above 150 °C, radicals are generated
that are amplified by PERCA. Since $O_3$ is typically present at mixing ratios in the tens of ppbv in ambient air, quantification of ΣPAN with an ethane-based TD-PERCA-CRDS would be challenging. In contrast, TD-PERCA instruments using CO will not have this limitation, as CO reacts with O to $CO_2$ and would not generate $RO_x$ radicals.

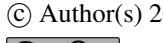



### 3.6.2 Interference from peroxyacetic acid in the measurement of ΣPAN at 250 °C

In a previous paper (Taha et al., 2018), we reported interference from peroxyacetic acid (PAA) when the inlet was operated at 250 °C. However, the mixing ratios delivered on those experiments were well above what is expected in ambient air. Further, it is unclear what fraction of PAA dissociates at 250 °C, since the Arrhenius parameters for TD of PAA are uncertain (Table 3). Regardless of whose Arrhenius parameters are assumed, the temperature needed to dissociate 0.1% of PAA is greater than that needed to dissociate 99.9% of PAN (Table 3). In ambient

air, PAA is present at concentrations of up to one order of magnitude greater than that of PAN (Phillips et al., 2013). We estimate that under typical conditions, the interference from PAA dissociation is <1% and likely be completed prevented if the inlet is operated at a temperature below 250 °C.

### 3.6.3 Interference from ΣPAN in the measurement of ΣPN at 95 °C – 110 °C

   A temperature of 95°C (110 °C) is required to dissociate >90% (>99.9%) of PNA in the TD-PERCA inlet; at these

temperatures, ~12% (~39%) of PAN dissociates (Figure 4). Since [PAN] > [PNA] (Table 1) and hence ΣPAN > ΣPN in most environments, the contribution of ΣPAN to the ΣPN signal in ambient air is substantial (and likely also variable given the slope of the PAN TD curve in this region). Hence, measurement of ΣPN in ambient air by TD-PERCA-CRDS with a single channel relative to an $NO_2$ background measurement is unfeasible.

### 3.7 Differential temperature TD-PERCA-CRDS for measurement of ΣPN and ΣPAN

**3.7.1 Synthetic air mixtures**

   To overcome the limitations outlined in section 3.6, a multichannel, differential temperature approach was used. Here, two channels were operated at constant temperatures set in the 60 °C to 110 °C range to avoid the interference from $O_3$ dissociation and ensure that response for ΣPAN remains linear (by dissociating only a fraction of its concentration). Since the amount of $NO_2$ generated by TD-PERCA is a function of temperature and radical chain

length ($CL_T$) as well as a fraction dissociated ($F_T$) of PNA and ΣPAN, the responses in the two PERCA channels operated at different temperatures, T1 and T2, are:

$$[NO_2]_{T1} = CL_{T1} \times F_{PAN,T1} \times [\Sigma PAN] + CL_{T1} \times F_{PNA,T1} \times [\Sigma PN] + [NO_2]_{ref} \qquad (1a)$$

$$[NO_2]_{T2} = CL_{T2} \times F_{PAN,T2} \times [\Sigma PAN] + CL_{T2} \times F_{PNA,T2} \times [\Sigma PN] + [NO_2]_{ref} \qquad (1b)$$

   If the $CL_T$ and $F_T$ values are measured at temperatures T1 and T2 independently (i.e., offline), and $[NO_2]_{ref}$ is

quantified in an unheated, parallel reference channel, the mixing ratios of ΣPN and ΣPAN can be calculated by rearranging equations (1a) and (1b):



$$[\Sigma PAN] = \frac{([NO_2]_{T1}-[NO_2]_{ref})\times CL_{T2}F_{PNA,T2}-([NO_2]_{T2}-[NO_2]_{ref})\times CL_{T1}F_{PNA,T1}}{CL_{T2}F_{PNA,T2}\times CL_{T1}F_{PAN,T1}-CL_{T2}F_{PAN,T2}\times CL_{T1}F_{PNA,T1}} \qquad (2a)$$

$$[\Sigma PN] = \frac{([NO_2]_{T2}-[NO_2]_{ref})-CL_{T2}F_{PAN,T2}\times[\Sigma PAN]}{CL_{T2}F_{PNA,T2}} \qquad (2b)$$

A time series demonstrating this approach using the four channel setup is presented in Figure 12A. Here, a constant
mixing ratio of PNA (along with $NO_2$ and $O_3$ from the photolysis source) was added to the inlet between 21:44 and
21:55. At 21:48:30 and at 21:50:45, PAN was added via the GC pre-column. The product of CL and F for PAN
and PNA at 110 °C, 80 °C, and 60 °C was determined offline and are summarized in Table 4 and assumed for same
for all $\Sigma PN$ and $\Sigma PAN$ species.

The time series of PNA and PAN mixing ratios derived from equations 2a and 2b are presented in Figure 12B.
Consistent results were obtained independent of which pair of channels was used in the calculations (Table 5).

### 3.7.2 Laboratory air

The differential temperature TD-PERCA-CRDS was then used to sample and determine $\Sigma PN$ and $\Sigma PAN$ in
laboratory air. The calibration parameters summarized in Table 4 were used since they were determined with
scrubbed air, which has the same RH as the air intake (i.e., the laboratory). The results are summarized in Table 6.
The calculated room air $\Sigma PN$ values mixing ratios are negative (i.e., not physically possible). On the other hand,
the $\Sigma PAN$ mixing ratios are unreasonably large as we have never observed similarly high mixing ratios in our
laboratory by GC, TD-CRDS, or by CIMS. Taken together, these observations suggest that there is (or are) species
that dissociate in the TD inlet and generate $RO_x$ radicals or, perhaps, atomic oxygen other than $\Sigma PAN$. These
unknown interfering species seem to have different TD profiles than PAN and PNA as the differential response
differs when different pairs of channels are used in the calculation. Furthermore, the response to the unknown
species is more prominent when the hottest (110 °C) channel is used in the calculation.

When ~260 pptv of PNA and ~480 pptv PAN (using the same setup as for Figure 12 and Table 5) were added to
the sampled laboratory air, the responses (i.e., $\Delta(\Sigma PN)$ and $\Delta(\Sigma PAN)$) are consistent in all channels, which suggests
that the chain lengths and dissociated fractions had not changed. This observation corroborates that the ethane
based TD-PERCA-CRDS in reality quantifies $\Sigma PAN^*$, which includes PAN, PPN, etc. plus one or more
unidentified species. In section 4, we speculate as to the potential identity of the interfering species.

### 3.8 Figures of merit

The ability of TD-PERCA-CRDS to detect radicals is limited by the instrument's ability to detect differences in
$NO_2$ concentration after amplification, calculated using (Brown et al., 2002):




$$[NO_2]_{min} = \frac{R_L}{c\sigma}\left(\frac{\Delta\tau_{min}}{\tau_0^2}\right) \qquad (3)$$

Here, $[NO_2]_{min}$ is the smallest $NO_2$ concentration that can be detected, $\Delta\tau_{min}$ is the smallest measurable difference between ring-down time constants in the presence ($\tau$) and absence ($\tau_0$) of $NO_2$, c is the speed of light, $\sigma$ is the $NO_2$ absorption cross-section at 405 nm ($6.1\times10^{19}$ $cm^2$ molecule$^{-1}$ (Paul and Osthoff, 2010)), and $R_L$ is a correction factor. $\Delta\tau_{min}$ is approximately (Brown et al., 2002):

$\Delta\tau_{min} = \sqrt{2}\times\sigma(\tau_0) \qquad (4)$

When sampling zero air, the LOD for $NO_2$ was 49 pptv (1 s, 1 $\sigma$). In the presence of NO and ethane reagent gases, the LOD increases to 87 pptv (1 s, 1 $\sigma$).

Employing the dual-channel-TD-PERCA-CRDS with the PERCA heater at 250 °C, a CL of 69 factored in, and in the absence of $NO_2$, the 1 s, 1 $\sigma$ LOD for $\Sigma PAN^*$ was 1.3 pptv. At an inlet temperature of 120 °C and with a CL

of 26, the LOD for PNA was 3.4 pptv. When averaging for 90 s, the minimum of an Allan variance plot (Figure 7 of Paul and Osthoff (2010)) the LOD improves to 8.7 pptv for $NO_2$ , 0.13 pptv for $\Sigma PAN^*$, and 0.33 pptv for PNA. The accuracy of TD-PERCA-CRDS is limited by uncertainties in CL ($\pm7\%$ for dry air), variability in the response to different type of peroxy radicals ($\pm9\%$) (Wood et al., 2016), and of the $NO_2$ measurement ($\pm4\%$) which is dominated by uncertainties in the absorption cross-section and $R_L$ (Paul and Osthoff, 2010). Adding these in

quadrature gives a combined uncertainty of $\pm12\%$ for dry air.

An additional uncertainty factor for the differential temperature TD-PERCA-CRDS is the uncertainty in $F_T$. The chambers are operated at temperatures where $F_T$ is highly sensitive to temperature (Figure 4). Judging from the scatter observed (for example, in Table 5), we estimate that an additional $\pm5\%$ random error is introduced, raising the combined measurement uncertainty to $\pm13\%$. Not included in this estimate are systematic errors that might

arise from the unknown and potentially variable TD profile of the interferences included in $\Sigma PAN^*$.

## 4 Discussion

The main goal of this work was to evaluate the feasibility of using ethane-based TD-PERCA to quantify $\Sigma PN$ and $\Sigma PAN$ in ambient air. This work has identified several stumbling blocks that on aggregate insinuate that such a measurement would be difficult and error-prone in practice.

On the one hand, the ethane-based TD-PERCA-CRDS has demonstrated great LODs (<1 pptv for $\Sigma PAN^*$ and PNA). This constitutes a considerable improvement compared to our previous generation TD-CRDS, whose LOD





was in the 100s of pptv (Paul and Osthoff, 2010), and represents the first optical absorption measurement of PNA at concentration levels of the same magnitude as found in ambient air (Table 1). In addition, the measurement can tolerate a large $NO_2$ background through selective amplification of the desired signal: In Figure 12, for example,

the $NO_2$ background was >30 ppbv, yet PNA and PAN were quantified with a 1 s, 1 σ precision of <6 pptv and <40 pptv, respectively (Table 5). Moreover, the sensitivity of the ethane-based TD-PERCA is better than the room temperature measurement of $RO_x$ radicals (1.6 pptv; 90 s, 2 σ) (Wood et al., 2016), mainly because of the greater amplification (~69 vs. ~25) and in spite of the CAPS sensor being slightly more sensitive to $NO_2$ than our CRDS. Furthermore, the instrument's sensitivity is comparable to (or better than) what is achievable with commonly used

GC and CIMS methods. For example, at an inline temperature of 120 °C, the sensitivity of TD-PERCA-CRDS for PNA was of the same order of magnitude as our CIMS at its non-specific ion at $m/z$ 62 and the optimized CIMS recently described by Chen et al. (2017).

The TD-PERCA-CRDS owes its good sensitivity to its high CL, which increases with temperature. Our attempts to rationalize the temperature through model simulations (see S.I.) were limited because models simulating PERCA

need to take wall loss rates into account and are generally poor predictors of experimental chain lengths. From a gas-phase kinetics perspective, reactions of $HO_2$ and $RO_2$ with NO (e.g., R3, Table 2) have a negative activation energy and are thus expected to slow down at higher temperatures, decreasing turnover rates and the CL. On the over hand, the RH dependence is reduced by heating, in part because one of the radical chain-terminating reaction, $HO_2 + NO \rightarrow HNO_3$, proceeds via a water adduct ($HO_2 \cdot H_2O$) (Butkovskaya et al., 2007; Butkovskaya et al., 2009).

The temperatures within the PERCA reaction heater are sufficiently elevated to dissociate this intermediate, shutting down this radical sink reaction. In addition, the elevated temperatures inside the reactor may lessen reactions at the reactor inner wall surfaces (by driving off adsorbed water molecules, for example) though we lack direct evidence for this happening.

On the other hand, however, the TD-PERCA-CRDS method has several drawbacks, some of which still need to be

overcome to make ambient measurements a reality.

The first challenge is posed by the TD profiles of PAN and PNA (Figure 4) which are not completely separated. This overlap is particularly problematic in ambient air because the signal generated by the typically much smaller PNA concentrations could be overshadowed by a much larger ΣPAN signal. In this work, the overlap of the TD profiles of PNA and PAN (and, the rather limited dynamic range of <500 pptv) was overcome by the differential

temperature / linear combination method (section 3.7) in which ΣPAN was only partially dissociated and PNA close to completely dissociated.



A complication is that methyl and ethyl peroxy nitrate have TD profiles that are similar, but not identical, to that of PNA; these molecules dissociate at lower temperatures than PNA (Table 3). This does not matter if the TD-PERCA inlets are operated at temperatures at which all three molecules are fully dissociated (or nearly so) as in
this work.

The differential temperature approach has the additional advantage of avoiding the $O_3$ interference that occurs above 150 °C, which would otherwise have been a serious issue because of the typically much larger $O_3$ than PAN or PNA concentrations in ambient air, and gave consistent results in synthesized air mixtures and room air.

A second drawback of TD-PERCA is the RH dependent CL, which necessitates frequent calibrations to determine
CL×F at each channel's temperature, though this could in principle be straightforward with photochemical sources of PNA and PAN and automated switching. At the same time, frequent calibrations of CL and F would relax the need to carefully match the responses in each of the TD channels, which is a requirement in conventional TD instruments.

A third and most substantial drawback are the interferences. The $O_3$ interference is easily avoided by switching the
instrument from ethane to CO operation. With CO, any O generated from $O_3$ dissociation would react to form $CO_2$ and be of no further consequence. However, CO-PERCA is somewhat unappealing because of CO's high toxicity and lack of smell, making its use impractical and impermissible in some university laboratories due to legitimate safety concerns. Mihele and Hastie (2000) used PAN as a radical source by heating a short section of the inlet of their CO-PERCA to 200 °C and found the CL to be the same as other radical sources (which included passing $H_2$
past a heated wire and a $Cl_2$ photolysis source), which suggests that the CL in a CO based PERCA is less dependent on temperature than with an ethane based PERCA, which would be another advantage of CO based PERCA.

The unknown interference observed in laboratory air is by far the biggest hurdle. We do not know the identity of the molecule or class of molecules interfering, and if the interference is present in ambient air, away from anthropogenic sources, or found only within a chemistry laboratory. We previously reported (Taha et al., 2018)
that the ethane-based TD-PERCA-CRDS responds to peroxide explosives and also responds when sampling the head space above common skin cream and moisturizer products, which contain organosiloxanes. Organosiloxanes have become ubiquitous in the environment and have been found in indoor air in ppbv levels (Rücker and Kümmerer, 2015). Their barriers to dissociation have been reported (Davidson and Thompson, 1971) and are too large for decomposition to occur at the inlet temperatures used in this work. However, Kulyk et al. (2016) recently
suggested that pyrolysis of certain polysiloxanes may occur at temperatures as low as 70 °C. Clearly, more work is needed to identify which molecules or class of molecules interferes in TD-PERCA-CRDS and are included in ΣPAN*. It is possible that the interfering species thermally dissociate(s) to release O atoms; if that's the case, this





interference would not be present in a CO based TD-PERCA. Hence, measurements using a CO-based TD-PERCA should be attempted.






**Acknowledgments**

This work was made possible by the financial support of the National Science and Engineering Research Council
of Canada (NSERC) in the form of a Discovery grant. The authors thank Ezra Wood for sharing a preprint version
of a manuscript and useful discussions. YMT, CZY, and NMG acknowledge financial support from the NSERC
Collaborative Research and Training Experience Program (CREATE) "Integrating Atmospheric Chemistry and
Physics from Earth to Space" (IACPES) and QEII graduate scholarships.





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





**Tables**

**Table 1. Typical\* PNA and PAN mixing ratios in various environments where both were quantified.**

| Location | PNA (pptv) | PAN (pptv) | PNA/PAN |
|---|---|---|---|
| Antarctica - Summer | 20 (Slusher et al., 2001) | 15.6 (Eisele et al., 2008) | 1.3 |
| Antarctica - Summer | 22 (Jones et al., 2014) | 15.6 (Eisele et al., 2008) | 1.4 |
| Antarctica – Summer | 2.5 (Jones et al., 2014) | 9.2 (Mills et al., 2007) | 0.27 |
| Remote troposphere – Spring | - | - | 0.13\*\* (Murphy et al., 2004) |
| Free troposphere (Intex-NA) - Summer | 45 (Kim et al., 2007) | 280 (Singh et al., 2006) | 0.16\*\*\* |
| Uintah Basin – Winter 2013 | 500 (Veres et al., 2015) | 2000 (Ahmadov et al., 2015) | 0.25 |
| Uintah Basin – Winter 2014 | 100 (Veres et al., 2015) | 300 (Lee et al., 2015) | 0.33 |
| Atlanta – Winter | 3.7 (Chen et al., 2017) | 640-800 (Lee et al., 2013) | <0.01 |
| Atlanta – Summer | 11.7 (Chen et al., 2017) | 640-800 (Lee et al., 2013) | 0.01-0.02 |

\* Average values

\*\* calculated assuming all non-PAN and PPN peroxy nitrate is PNA.

\*\*\* Averages of data posted on ftp://ftp-air.larc.nasa.gov/pub/INTEXA/DC8_AIRCRAFT/



**Table 2. List of chemical reactions[*]**

| Number | Reaction | k (cm$^3$ molecule$^{-1}$ s$^{-1}$) | k$_{298K}$ (cm$^3$ molecule$^{-1}$ s$^{-1}$) |
|---|---|---|---|
| R1 | $CH_3O_2 + NO \rightarrow CH_3O + NO_2$ | $2.3 \times 10^{-12} \times e^{(360/T)}$ | $7.7 \times 10^{-12}$ |
| R2 | $CH_3O + O_2 \rightarrow HCHO + HO_2$ | $7.2 \times 10^{-14} \times e^{(-1080/T)}$ | $1.9 \times 10^{-15}$ |
| R3 | $HO_2 + NO \rightarrow HO + NO_2$ | $3.5 \times 10^{-12} \times e^{(270/T)}$ | $8.5 \times 10^{-12}$ |
| R4 | $C_2H_6 + HO + O_2 \rightarrow H_2O + C_2H_5O_2$ | $6.9 \times 10^{-12} \times e^{(-1000/T)}$ | $2.4 \times 10^{-13}$ |
| R5 | $C_2H_5O_2 + NO \rightarrow C_2H_5O + NO_2$ | $2.55 \times 10^{-12} \times e^{(380/T)} \times 0.99$ | $9.1 \times 10^{-12}$ |
| R6 | $C_2H_5O + O_2 \rightarrow CH_3CHO + HO_2$ | $2.4 \times 10^{-14} \times e^{(-325/T)}$ | $8.1 \times 10^{-15}$ |
| R7 | $PAN \rightarrow CH_3CO_3 + NO_2$ | See Table 3 | $4.4 \times 10^{-4}$ s$^{-1}$ |
| R8 | $CH_3C(O)O_2 + NO \rightarrow NO_2 + CH_3C(O)O \rightarrow NO_2 + CH_3 + CO_2$ | $7.5 \times 10^{-12} \times e^{(290/T)}$ | $2.0 \times 10^{-11}$ |
| R9 | $PPN \rightarrow C_2H_5CO_3 + NO_2$ | See Table 3 | $3.7 \times 10^{-4}$ s$^{-1}$ |
| R10 | $C_2H_5CO_3 + NO + O_2 \rightarrow C_2H_5O_2 + NO_2 + CO_2$ | $6.7 \times 10^{-12} \times e^{(340/T)}$ | $2.1 \times 10^{-11}$ |
| R11 | $HO + NO + M \rightarrow HONO + M$ | $7.4 \times 10^{-31} \times (T/300)^{-2.4} \times [M]$ | $9.7 \times 10^{-12}$ [**] |
| R12 | $HO_2 + NO_2 + M \rightarrow HO_2NO_2 + M$ | (termolecular) | (termolecular) |
| R13 | $HO_2 + HO_2 + M \rightarrow H_2O_2 + M$ | (termolecular) | (termolecular) |
| R14 | $CH_3CO_3 \rightarrow 0.7\ CH_3CO_2 + 0.3\ CH_3CO_2H$ | $5.0 \times 10^{-12} \times \sum RO_2$ | $5.0 \times 10^{-12} \times \sum RO_2$ |
| R15 | $C_2H_5CO_3 \rightarrow 0.7\ C_2H_5CO_2 + 0.3\ C_2H_5CO_2H$ | $5.0 \times 10^{-12} \times \sum RO_2$ | $5.0 \times 10^{-12} \times \sum RO_2$ |
| R16 | $CH_3O_2 \rightarrow 0.330\ CH_3O + 0.335\ HCHO + 0.335\ CH_3OH$ | $1.8 \times 10^{-13} \times e^{(416/T)} \times \sum RO_2$ | $7.4 \times 10^{-13} \times \sum RO_2$ |
| R17 | $C_2H_5O_2 \rightarrow 0.6\ C_2H_5O + 0.2\ CH_3CHO + 0.2\ C_2H_5OH$ | $3.1 \times 10^{-13} \times \sum RO_2$ | $3.1 \times 10^{-13} \times \sum RO_2$ |

[*] Rate constant expressions are from the Master Chemical Mechanism (MCM),(Jenkin et al., 1997; Saunders et al.,

2003) version 3.3.1, except R7 and R9, which are from Kabir et al. (2014)

[**] calculated using $[O_2] = 4.2 \times 10^{18}$ molecules cm$^{-3}$



**Table 3: Arrhenius parameters for thermal dissociation of selected PN, PAN, PPN, PAA, and O₃.**

| Molecule | A (s⁻¹) | $E_a$ (kJ mol⁻¹) | Reference | T needed to dissociate 0.1% (°C)** | T needed to dissociate 99.9% (°C)** |
|---|---|---|---|---|---|
| $HO_2NO_2$ (PNA) | $7.3 \times 10^{14}$ * | $88.1 \pm 4.4$ | (Atkinson et al., 1997) | 24 | 123 |
| $CH_3O_2NO_2$ (MPN) | $1.1 \times 10^{16}$ | $88.1 \pm 4.4$ | (Atkinson et al., 1997) | 3 | 86 |
| $C_2H_5O_2NO_2$ (EPN) | $8.8 \times 10^{15}$ | $86.5 \pm 8.7$ | (Atkinson et al., 1997) | 0 | 82 |
| $CH_3C(O)O_2NO_2$ (PAN) | $2.8 \times 10^{16}$ | $113 \pm 2$ | (Kabir et al., 2014) | 73 | 174 |
| $C_2H_5C(O)O_2NO_2$ (PPN) | $2.36 \times 10^{16}$ | $113 \pm 2$ | (Kabir et al., 2014) | 75 | 176 |
| $CH_3C(O)O_2H$ (PAA) | $10^{14}$ | $134 \pm 8$ | (Schmidt and Sehon, 1963) | 206 | 377 |
| $CH_3C(O)O_2H$ (PAA) | $1.15 \times 10^{13}$ | 136 | (Devush et al., 1983) | 247 | 450 |
| $CH_3C(O)O_2H$ (PAA) | $5 \times 10^{14}$ | $168 \pm 4$ | (Sahetchian et al., 1992) | 300 | 492 |
| $O_3$ | $1.3 \times 10^{10}$ * | 92.8 | (Jones and Davidson, 1962; Heimerl and Coffee, 1979) | 180 | 433 |

* Calculated assuming a pressure of 550 Torr and temperature of 298 K.

** Assuming a contact time of 4 ms at the maximum temperature (Paul et al., 2009)

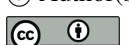



**Table 4: Products of CL and F for PNA and PAN at 110 °C, 80 °C, and 60 °C (RH = 34%)**

|       | $CL_{110\ °C} \times F_{110\ °C}$ | $CL_{80\ °C} \times F_{80\ °C}$ | $CL_{60\ °C} \times F_{60\ °C}$ |
|-------|-----------------------------------|----------------------------------|----------------------------------|
| PNA   | $41.8 \pm 0.2$                    | $39.5 \pm 0.2$                   | $31.9 \pm 0.1$                   |
| ΣPAN  | $10 \pm 1$                        | $7.6 \pm 0.9$                    | $1.2 \pm 0.2$                    |

**Table 5: Statistics (average ± 1 standard deviation) of the 1 s data shown in Figure 12.**

|                         | PNA (pptv)                          |                                     |                                    | PAN (pptv)                          |                                     |                                    |
|-------------------------|-------------------------------------|-------------------------------------|------------------------------------|-------------------------------------|-------------------------------------|------------------------------------|
| Time period             | T1 = 110 °C<br>T2 = 60 °C           | T1 = 110 °C<br>T2 = 80 °C           | T1 = 80 °C<br>T2 = 60 °C           | T1 = 110 °C<br>T2 = 60 °C           | T1 = 110 °C<br>T2 = 80 °C           | T1 = 80 °C<br>T2 = 60 °C           |
| 21:48:45 –<br>21:49:15  | $266 \pm 4$                         | $254 \pm 17$                        | $267 \pm 4$                        | $447 \pm 26$                        | $495 \pm 75$                        | $430 \pm 31$                       |
| 21:51:15 –<br>21:52:45  | $261 \pm 5$                         | $258 \pm 14$                        | $261 \pm 6$                        | $480 \pm 32$                        | $491 \pm 68$                        | $476 \pm 38$                       |



**Table 6: Statistics (average ± 1 standard deviation) of the 1 s data of laboratory air (first time period), laboratory air spiked with ~260 pptv PNA (second period), and laboratory air spiked with 260 pptv PNA and ~480 pptv PAN (third period). n/a = not applicable.**

| Time period | ΣPN (ppbv) | | | ΣPAN (ppbv) | | |
|---|---|---|---|---|---|---|
| | T1 = 110 °C T2 = 60 °C | T1 = 110 °C T2 = 80 °C | T1 = 80 °C T2 = 60 °C | T1 = 110 °C T2 = 60 °C | T1 = 110 °C T2 = 80 °C | T1 = 80 °C T2 = 60 °C |
| Room air | -0.08 ± 0.01 | -0.29 ± 0.03 | -0.068 ± 0.004 | 4.21 ± 0.04 | 5.1 ± 0.1 | 3.89 ± 0.03 |
| Room air + PNA | 0.19 ± 0.01 | -0.03 ± 0.02 | 0.20 ± 0.01 | 3.50 ± 0.03 | 4.4 ± 0.1 | 3.17 ± 0.04 |
| Δ(ΣPN) | 0.27 ± 0.01 | 0.26 ± 0.04 | 0.27 ± 0.01 | n/a | n/a | n/a |
| Room air + PNA + PAN | 0.19 ± 0.01 | -0.02 ± 0.02 | 0.20 ± 0.01 | 3.99 ± 0.04 | 4.8 ± 0.1 | 3.70 ± 0.03 |
| Δ(ΣPAN) | n/a | n/a | n/a | 0.49 ± 0.04 | 0.40 ± 0.14 | 0.52 ± 0.05 |



**Figures**

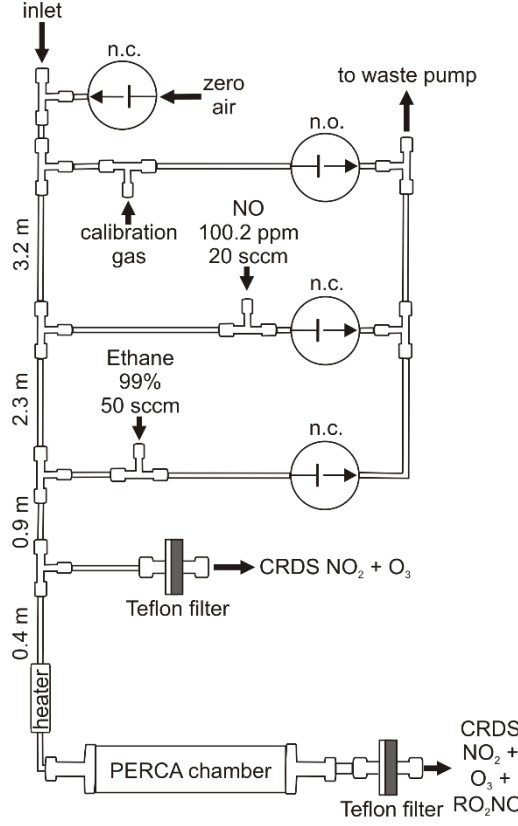

**Figure 1. Schematic of the dual-channel thermal decomposition peroxy radical chemical amplification inlet.**

**The inlet consists of a 60 cm long, 0.635 cm (¼") o.d. quartz heater and an 80 cm long, 1.27 cm (½") o.d.**

**FEP Teflon™ reaction chamber. Reaction gases were added upstream of the PERCA chamber by closing**

**the normally open valves connected to a waste pump via 50 μm critical orifices. Background NO₂ levels were**

**monitored in a parallel detection channel by tapping into the inlet prior to thermal dissociation of peroxy**

**nitrates.**

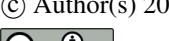



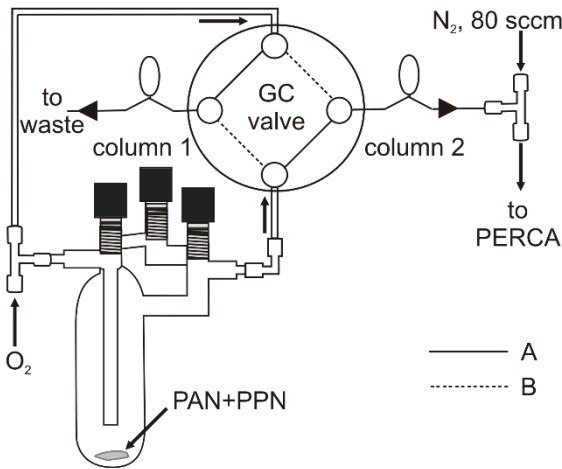

**Figure 2. Setup for delivery of PAN and PPN using a gas chromatography column.**


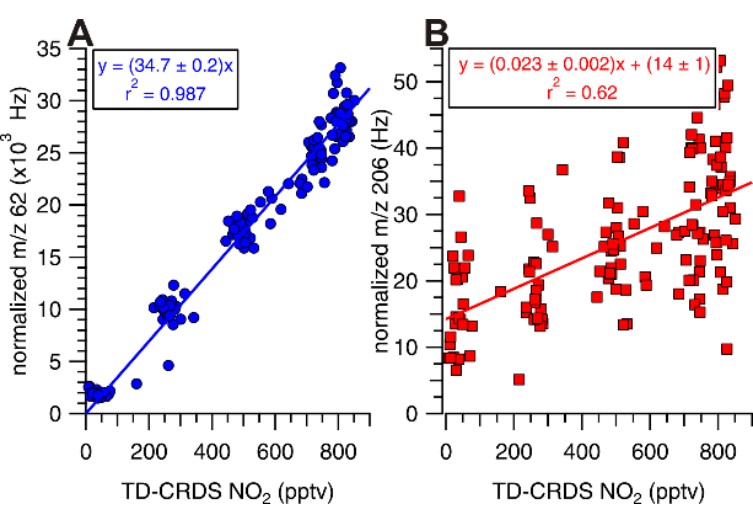

**Figure 3. Calibration of CIMS response factors against TD-CRDS operated with its inlet heated to 120 °C**

**and without amplification. (A)** *m/z* **62. (B)** *m/z* **206. CIMS counts were normalized to $10^6$ I⁻ counts.**






**Figure 4. Normalized thermal dissociation profiles of PNA, PAN, PPN, and O₃ as a function of inlet set temperature. Superimposed trend lines are simulations based on the TD model introduced by Paul et al. (2009). The PNA, PAN, and PPN data were observed by TD-CRDS without amplification gases present, whereas the O₃ data were observed by TD-PERCA-CRDS. The error bars represent standard deviations of 1 s data.**





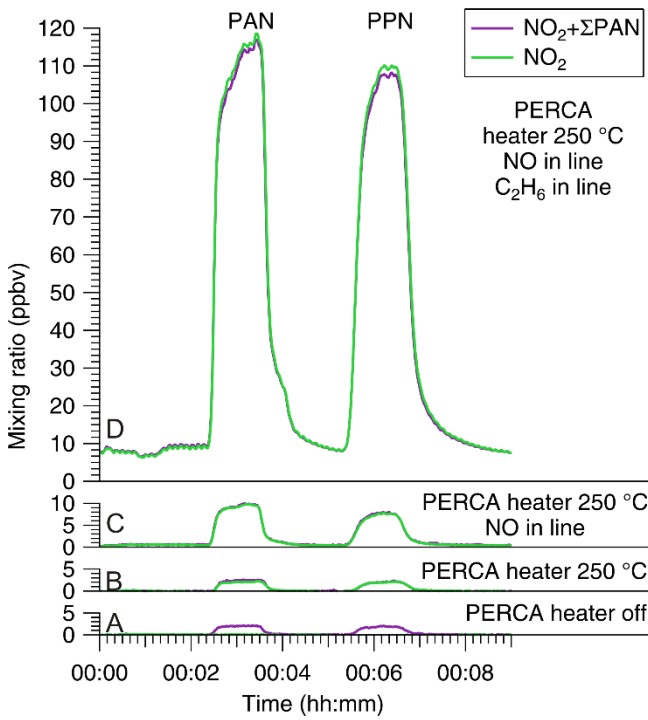

**Figure 5. Peroxy radical chemical amplification of peroxyacetic and peroxypropionic nitric anhydride (PAN and PPN) delivered via a megabore GC column. (A) Time series of the signal observed by cavity ring-down spectroscopy in the ambient temperature (NO$_2$; green colour) and heated (NO$_2$ + $\sum$PAN; purple colour) channels with the PERCA heater off. (B) Same as (A) with the PERCA heater switched on. (C) Same as (B) with 0.75 ppmv NO added. (D) Same as (C) with 1.5% C$_2$H$_6$ added. The amplification factor is determined from the ratio of the ambient temperature (i.e., NO$_2$) CRDS signal observed in (D) divided by that observed in (B).**






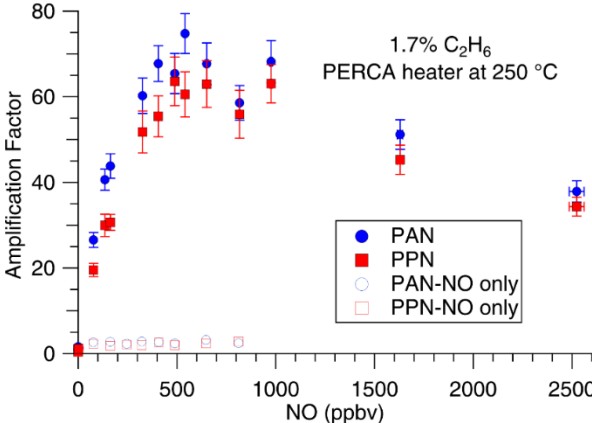

**Figure 6. TD-PERCA-CRDS amplification factors of ~0.5 ppbv PAN and ~1.3 ppbv PPN as a function of**

**NO mixing ratio at RH = 0%. The error bars represent standard deviations of 1 s data.**






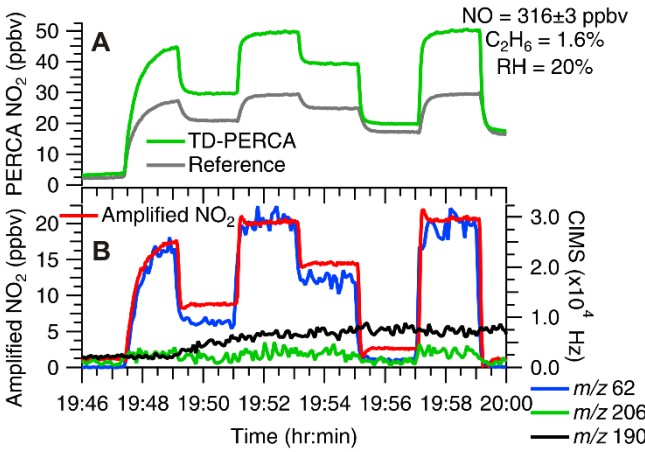

**Figure 7. (A) Sample time series of PNA observed by TD-PERCA-CRDS in the reference, NO₂ channel (shown in green) and PERCA channel (grey). (B) (left) Difference signal between amplified and reference channel (shown in red). (right) CIMS counts (normalized to $10^6$ I⁻) at *m/z* 62, the major fragment (NO₃⁻) expected from PNA, at *m/z* 206 (multiplied by a factor of 100 for clarity), the HNO₄·I⁻ cluster, and at m/z 190, the HNO₃·I⁻ cluster.**




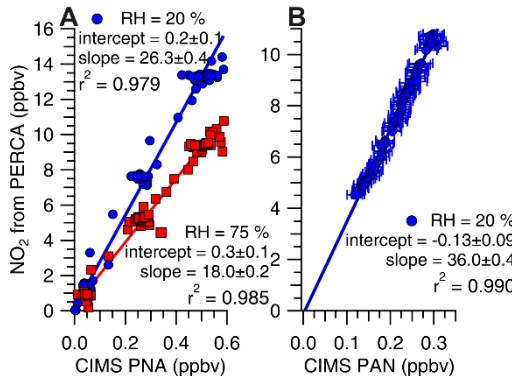

**Figure 8. Scatter plots of TD-PERCA-CRDS and CIMS measurements. (A) Sample PNA calibrations at an NO mixing ratio of 316±3 ppbv and TD-PERCA inlet temperature of 120 °C (B) Sample PAN calibration at an NO mixing ratio of 662±2 ppbv and TD-PERCA inlet temperature of 250 °C.**






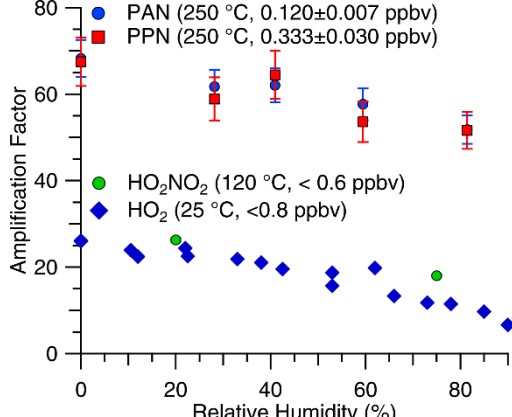

**Figure 9. TD-PERCA-CRDS amplification factor as a function of relative humidity at an ethane mixing ratio of 1.6% and with NO mixing ratios of 644±2 and 316±3 ppbv for the PAN/PPN experiments and the PNA experiments, respectively. The PAN and PPN mixing ratios were 0.120±0.007 and 0.333±0.030 ppbv, respectively. The PNA mixing ratio was varied between 0 and 0.6 ppbv. The room temperature HO₂ data are from Wood et al. (2016).**



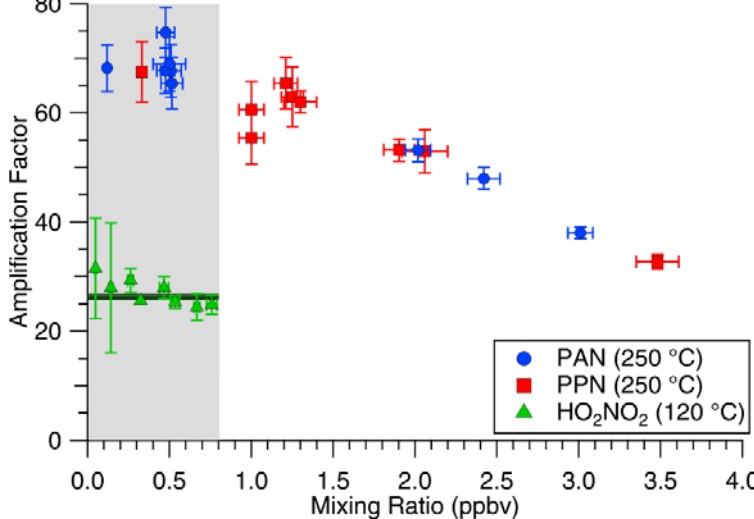

**Figure 10. TD-PERCA-CRDS amplification factors as a function of PAN, PPN, and PNA mixing ratio.**

**Errors bars correspond to ±1σ standard deviation. Mixing ratios of NO and ethane were 500±50 ppbv and**

**~1.6%, respectively. The grey underlay indicates the linear range. The dark green line corresponds to**

**26.3±0.4 (slope of data shown in Figure 8A).**




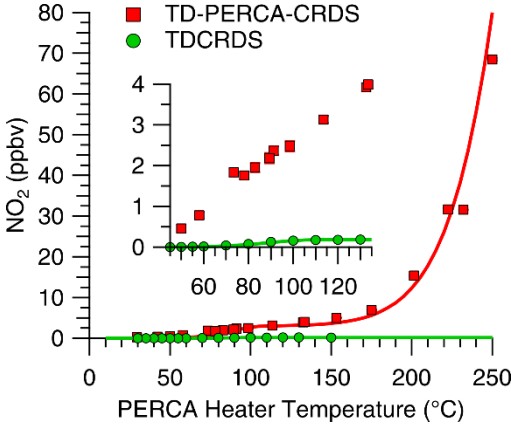

**Figure 11. Amplification of photochemically generated PNA (red) as a function of PERCA heater temperature (RH=24%). Simultaneous quantification by CIMS and UV absorption showed that the gas stream contained 180±28 pptv PNA and 3.5±0.2 ppbv O₃. The non-amplified TD-CRDS signal (from Figure**

**4), multiplied by 0.18, is shown in green for comparison.**






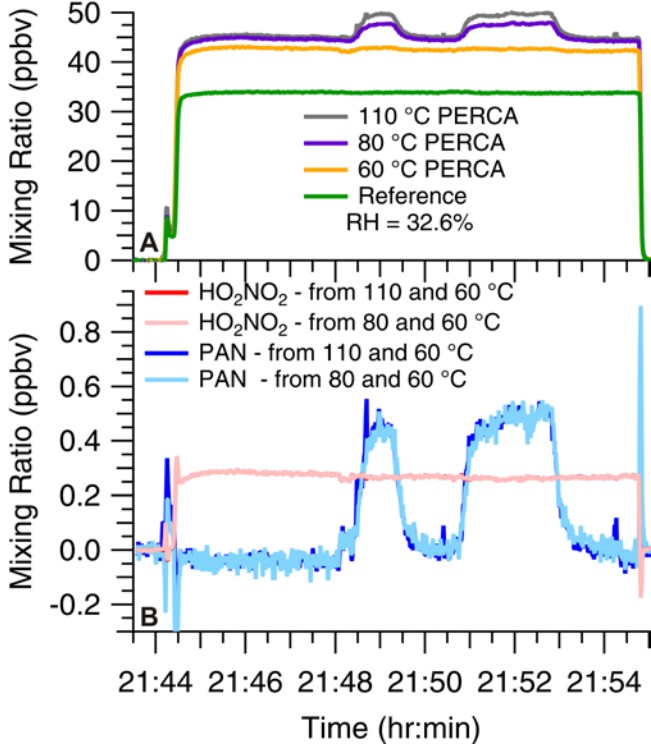

**Figure 12. Demonstration of differential temperature TD-PERCA-CRDS. (A) Time series of NO₂ mixing ratios observed by an unheated reference channel and three TD-PERCA channels operated at 60 °C, 80 °C, and 110 °C, respectively. PNA was sampled from 21:44:30 to 21:54:45, and PAN was added at ~21:49 and ~21:52. (B) Mixing ratios of PAN and PNA calculated from (A).**
