# Peer review of "Quantification of peroxynitric acid and peroxyacyl nitrates using an ethane-based thermal dissociation peroxy radical chemical amplification cavity ring-down spectrometer"

_Atmospheric Measurement Techniques, 2018_

## Referee Comment (RC1) · Anonymous Referee #1 · 21 May 2018

Taha et al., report an approach to quantifying peroxynitric acid and peroxyacyl nitrates (PNs and PANs) by combining the techniques of differential thermal dissociation (TD), peroxy radical chemical amplification (PERCA) and cavity ring down spectrum for $NO_2$ detection (CRDS), named TD-PERCA-CRDS. The instrument has single channel (on-off mode), duel channel (background $NO_2$, background + amplified $NO_2$) and four channel detection modes (differential temperature measurement). The chemical amplification module is optimized by injecting 0.6 ppm NO and 1.6% $C_2H_6$, the chain length (CL) is range from 20 to 70. They found that the measurement was suffered with the ambient $O_3$ interferences by thermal dissociating $O_3$ to O above 150 $^oC$ and the following reaction with ethane; the differential temperature TD-PERCA-CRDS at 60-110 $^oC$ was also unfeasible in ambient due to the unknown interferences in ambient air. Compared with TD-CRDS /LIF/CAPS, this technique still can't be well performed in field measurement, while the adding of PERCA provide a way to improve the detecting capacity of PNs and PANs to several pptv level. Overall the manuscript is well written and would be of interest and contribute to the community, I recommend this paper to be published on AMT subject to these comments below.

Major comments:

1. This instrument can calibrate the CL for several PNs or PPN produced in the lab at certain temperature and RH, but the CL is highly varied for different kinds of peroxynitric acid and peroxyacyl nitrates in ambient conditions, which also prevent the application of field measurement.

2. Is it possible to carry out the simultaneous measurement of CIMS with the four channel detection measurement in ambient condition to look insight of the unknown species? CIMS measurement might provide some useful information about the unknown interference.

Specific comments:

1. Line 105, if the majority of the experiments conducted by single-channel TD-PERCA-CRDS, the schematic figure of the single channel is worth to be shown in

Figure 1.

2. Line 45 and line 457: "10s to a few 100s" change to "tens to a few hundreds of".

3. Line 228, it is not necessary to set the single sentence to a paragraph.

4. The format of tables (1-6) does not conform to the academic norm.

5. Section 3.5.2, suggest the authors unify all the units to be pptv.

6. Line 893 the legend in Figure 7. (A): "Legend Sample time series of PNA observed by TD-PERCA-CRDS in the reference, $NO_2$ channel (shown in **green**) and PERCA channel (**grey**)." is not consist with the description in line 274, please correct it.

7. Is figure 8(A) only show part of the data presented in figure 7(A), as we can see the maximum $NO_2$ in figure 7(A) was about 20 ppbv while in figure 8(A) the maximum was only 16 ppbv. The authors should clarify it.

8. Figure 10 shows the CL dependence on the radical concentration (PAN, PPN and PNA), the PNA seems decreasing with the increasing of PNA concentration, and in line 320-335, no text gave the description or explanation about PNA dependence, I suggest adding some words to describe the CL dependence on the PNA concentration.

9. I suggest the authors summarizing the existed TD techniques that applied in the field measurement of peroxynitric acid or peroxyacyl nitrates, which offers convenience to the readers and contributes to the community.

---

## Referee Comment (RC2) · Anonymous Referee #2 · 5 Jun 2018

This paper describes laboratory characterization of a new method for quantifying peroxynitric acid (PNA) and peroxyacyl nitrates. Overall the article is well written and describes important results and I recommend publication after the mostly minor issues below are addressed.

Line 79: "The measurement of peroxy radicals by PERCA is prone to interferences", but the text proceeds to discuss that the amplification must be determined by calibrations and that it varies with relative humidities. These are not interferences! Later in the text an actual interference by ozone for TD-PERCA_CRDS is well described.... But

variation of calibration factor with RH is not an interference. Same for 89: replace "interference" with "disadvantage" or "property"?

Ling 143 – "Teflon" – what kind – PFA? PTFE?

Section 2.3.2 – clarify that the concentration of PNA is determined by the NO2 mixing ratio, correct? Ie, NO2 is the limiting reagent and HO2 is in excess. Line 190-191 – O2 is not readily photolyzed to form O3 by 254 nm – replace with "…generated by photolysis of O2 by 185 nm radiation from a low-pressure mercury lamp"?

Box model simulations (in SI) The SI discusses formation of C2H5ONO and C2H5O2NO2, but what about the temperature dependence of C2H5ONO2? That is, ethyl nitrate, formed by C2H5O2 + NO.

Section 3.2 and figure 5. This is overall very good demonstration of the technique. It is a bit confusing that, apparently, both the inlet heater and PERCA chamber can be heated separately. This should be more explicitly pointed out in the earlier experimental sections.

Ling 281: the text in the parenthesis, though likely true, makes the sentence awkward to read

Section 3.5: interestingly the amplification factor for PNA (yielding HO2) is less than that for PAN (which forms CH3CO3). The following sections address details of the chain length with T, RH, but is there is a conclusion for why the PNA vs. PAN results are so different?

Line 309: "…operated under optimal conditions and …" – this assumes that the optimal conditions do not change under varying circumstances. Might it be possible that the optimum NO or ethane concentrations are different at different temperatures or RH values?

SEction 3.7.2: The observed interferences are very interesting, and are likely relevant not only to TD-PERCA-CRDS but also to non-amplified thermal dissociation methods,

e.g. TD-LIF.

Section 3.8 , discussion of detection limit. Some of the terms here are confusing. 1. Do the authors actually mean precision when they have written LOD? LOD needs to be defined – is it for signal to noise ratio of 2? Or 3? The LOD is quoted as 87 ppt (1 sigma, 1 sec), but this seems much more like a description of the precision, not the LOD (ie, 1 sigma for precision, signal to noise ratio for LOD). 2. The authors have taken the "LOD" for the CRDS of 87ppt (1 sigma, 1 s) and divided by the CL of 69 to come up with the LOD for PANs of 1.3 ppt. Realistically, measuring PANs involves measuring NO2 twice - in amplification mode and in reference mode (either sequentially in a single channel instrument, or simultaneously with in a multi-channel instrument), so there should probably be another factor of sqrt(2). Also, the authors point out that the precision of the CRDS NO2 measurement is affected by the presence of NO and ethane reagent gases. For measurement in ambient air, or laboratory air, what is the precision of measuring NO2? The LOD (and precision) for an actual PNs measurement in ambient air would be affected by the precision of the CRDS NO2 measurement at the actual measurement conditions. For example, if O3 is 25 ppb, some portion of the O3 will react with the NO to give up to 25 ppb NO2 – is the precision the same at 0 ppb and 25 ppb? This has likely been addressed in earlier NO2 CRDS papers but should be mentioned for the reader's sake.
* * *

---

## Author Response (AR1)

We thank the reviewers for their time and effort reviewing this manuscript. All reviewer comments are reproduced below in **bold, italicized font**. Our responses are shown in regular font. Changes to the text are indicated as underlined text for insertions or are  for deletions. Line numbers given below are for the revised version with all markups shown. We numbered the reviewer comments for easier cross-referencing.

***Anonymous Referee #1***

***Taha et al., report an approach to quantifying peroxynitric acid and peroxyacyl nitrates (PNs and PANs) by combining the techniques of differential thermal dissociation (TD), peroxy radical chemical amplification (PERCA) and cavity ring down spectrum for NO2 detection (CRDS), named TD-PERCA-CRDS. The instrument has single channel (on-off mode), duel channel (background NO2, background + amplified NO2) and four channel detection modes (differential temperature measurement). The chemical amplification module is optimized by injecting 0.6 ppm NO and 1.6% C2H6, the chain length (CL) is range from 20 to 70. They found that the measurement was suffered with the ambient O3 interferences by thermal dissociating O3 to O above 150 oC and the following reaction with ethane; the differential temperature TD-PERCA-CRDS at 60110 oC was also unfeasible in ambient due to the unknown interferences in ambient air.***

***Compared with TD-CRDS /LIF/CAPS, this technique still can't be well performed in field measurement, while the adding of PERCA provide a way to improve the detecting capacity of PNs and PANs to several pptv level. Overall the manuscript is well written and would be of interest and contribute to the community, I recommend this paper to be published on AMT subject to these comments below.***

We thank the reviewer for these kind comments.

***Major comments:***

***1. This instrument can calibrate the CL for several PNs or PPN produced in the lab at certain temperature and RH, but the CL is highly varied for different kinds of peroxynitric acid and peroxyacyl nitrates in ambient conditions, which also prevent the application of field measurement.***

We agree that the variability of the chain lengths is a challenge and acknowledge that field measurements will require further work before they can become a reality (lines 492): "... the TD-PERCA-CRDS method has several drawbacks, some of which still need to be overcome to make ambient measurements a reality."

However, these are unsurmountable. Variable chain lengths also occur in regular PERCA instruments, and this has not prevented measurement of OH and $\Sigma RO_2$ with those instruments. Since the main chain carriers are OH and $HO_2$, and the chain lengths are large, the variability is not because of the nature of the radical precursor, but rather functions of temperature, humidity, and reagent gas concentrations. As we stated on line 507, frequent calibrations will be a necessity for this reason.

We have not amended the manuscript in response to this comment.

**2. Is it possible to carry out the simultaneous measurement of CIMS with the four channel detection measurement in ambient condition to look insight of the unknown species? CIMS measurement might provide some useful information about the unknown interference.**

This is an interesting suggestion. With CIMS operated with iodide reagent ion, it is certainly possible to quantify $HNO_4$, PAN, PPN, etc., and much could be learned from a side-by-side comparison. It could, for example, give insight into the magnitude and time-of-day dependence of the interference.

We have added to the following on line 532:

"Furthermore, such measurements should be complemented by parallel measurements of PNA, PAN and PPN by CIMS."

**Specific comments:**

**1. Line 105, if the majority of the experiments conducted by single-channel TDPERCA-CRDS, the schematic figure of the single channel is worth to be shown in Figure 1.**

This schematic is shown in the Electronic Supplementary Material of (Taha et al., 2018), which is subject to copyright. We have therefore chosen not to reproduce it here, but altered to the text on line 105 as follows:

"The majority of the experiments described in this manuscript were conducted using a single-channel TD-PERCA inlet that is described and depicted as Figure S1 of the Electronic Supplementary Material o (Taha et al., 2018)."

**2. Line 45 and line 457: "10s to a few 100s" change to "tens to a few hundreds of".**

We have changed the manuscript as requested by the reviewer.

**3. Line 228, it is not necessary to set the single sentence to a paragraph.**

This has been corrected.

**4. The format of tables (1-6) does not conform to the academic norm.**

We believe the reviewer is referring to the following instructions (taken from the Manuscript preparation guidelines for authors) on the AMT web site:

"Horizontal lines should normally only appear above and below the table, and as a separator between the head and the main body of the table. Vertical lines must be avoided."

We have amended the table borders as per the above instructions.

*5. Section 3.5.2, suggest the authors unify all the units to be pptv.*

We have changed the manuscript as suggested by the reviewer.

*6. Line 893 the legend in Figure 7. (A): "Legend Sample time series of PNA observed by TD-PERCA-CRDS in the reference, NO2 channel (shown in green) and PERCA channel (grey)." is not consist with the description in line 274, please correct it.*

We apologize - the colors were reversed in Figure 7A. $NO_2$ is now shown in green, and the heated channel in ash (aka grey) colour, consistent with the main text (lines 275-280) and the caption of Figure 7A.

*7. Is figure 8(A) only show part of the data presented in figure 7(A), as we can see the maximum NO2 in figure 7(A) was about 20 ppbv while in figure 8(A) the maximum was only 16 ppbv. The authors should clarify it.*

Figure 8A was set to <0.6 ppbv on the x-axis and <16 ppbv on the y axis, as this was the linear dynamic range. We updated the figure to show the entire concentration range and also moved the result of the linear fits (slope, intercept and $r^2$) to the figure caption for clarity.

*8. Figure 10 shows the CL dependence on the radical concentration (PAN, PPN and PNA), the PNA seems decreasing with the increasing of PNA concentration, and in line 320-335, no text gave the description or explanation about PNA dependence, I suggest adding some words to describe the CL dependence on the PNA concentration.*

We agree with the reviewer that the PNA data in Figure 10 at first glance look as if they decrease with concentration. However, looks are deceiving in this case since this trend falls within the scatter of the individual measurements, and Figure 8A clearly shows the absence of a concentration dependence. The main chain carriers are identical ($HO_2$ and OH) for PAN and PNA, such that one would expect the dynamic range for PNA to be the same as that for PAN or PPN, so PNA does not need to be discussed separately.

We modified a sentence on line 324 to make this clearer:

"It is well known in the PERCA community that the chain lengths decrease at high radical concentrations due to radical-radical reactions. Figures 8A and 8B demonstrate that the response of TD-PERCA-CRDS is linear for both PNA and PAN/PPN at low, atmospherically relevant mixing ratios (i.e., below ~600 pptv). The linear dynamic range is similar for PNA and PAN and PPN since the radical chain carriers are the same for both."

*9. I suggest the authors summarizing the existed TD techniques that applied in the field measurement of peroxynitric acid or peroxyacyl nitrates, which offers convenience to the readers and contributes to the community.*

We have added a Table summarizing TD techniques on line 783 as requested by the reviewer.

**Table 2. Selected thermal dissociation methods for quantification of daytime $NO_y$ species.**

| Species quantified | $NO_2$ detection method | Group | Reference |
|---|---|---|---|
| $NO_2$, ΣPAN, ΣAN, $HNO_3$ | LIF | Berkeley | (Day et al., 2002) |
| ΣPN | LIF | Berkeley | (Murphy et al., 2004) |
| HONO | CL | Berkeley | (Perez et al., 2007) |
| $NO_2$, ΣPAN, ΣAN | CRDS | Calgary | (Paul et al., 2009) |
| Aerosol nitrates | LIF | Berkeley | (Rollins et al., 2010) |
| $ClNO_2$ | CRDS | Calgary | (Thaler et al., 2011) |
| $NO_2$, ΣPAN, ΣAN, $HNO_3$ | LIF | L'Aquila | (Di Carlo et al., 2013) |
| NO, $NO_2$, HONO, $NO_y$, ammonium nitrate | CRDS | NOAA | (Wild et al., 2014; Womack et al., 2017) |
| $NO_2$, ΣPAN, ΣAN | CRDS | Max-Planck-Institut | (Thieser et al., 2016) |
| $NO_2$, ΣPAN, ΣAN | CAPS | Osaka | (Sadanaga et al., 2016) |
| $NO_2$, $RNO_2$ | CRDS | Hefei | (Chen et al., 2017b) |
| ΣPN, ΣPAN | PERCA-CRDS | Calgary | this work |

*Anonymous Referee #2*

*This paper describes laboratory characterization of a new method for quantifying peroxynitric acid (PNA) and peroxyacyl nitrates. Overall the article is well written and describes important results and I recommend publication after the mostly minor issues below are addressed.*

We also thank reviewer #2 for this assessment.

*Line 79: "The measurement of peroxy radicals by PERCA is prone to interferences", but the text proceeds to discuss that the amplification must be determined by calibrations and that it varies with relative humidities. These are not interferences! Later in the text an actual interference by ozone for TD-PERCA_CRDS is well described .... But variation of calibration factor with RH is not an interference.*

The reviewer is correct, of course. We modified the text on line 79 as follows:

"The measurement of peroxy radicals by PERCA is prone to matrix effects and interferences. For instance, a key operational parameter of any PERCA instrument is the radical chain length or amplification factor (CL), ..."

*Same for 89: replace "interference" with "disadvantage" or "property"?*

The thermal decomposition of PAN produces radicals and interferes in the measurement of ROx radicals by PERCA, so we believe that the word "interference" is used correctly in this context.

We modified the text on line 87 to improve its clarity:

"... apply heat. When quantification of ambient $RO_x$ radicals is the goal, this is avoided to prevent TD of ΣPN or ΣPAN (which are more abundant than free $RO_x$ radicals). TD of ΣPN or ΣPAN produces radicals that  interfere with the measurement of free $RO_x$ radicals (Mihele and Hastie, 2000). On the other hand, if measurement of ΣPN or ΣPAN is desired (such as in this paper), this interference is turned into a measurement principle. "

*Ling 143 – "Teflon" – what kind – PFA? PTFE?*

It is FEP. We inserted "fluorinated ethylene propylene (FEP)" prior to Teflon on line 145.

***Section 2.3.2 – clarify that the concentration of PNA is determined by the NO2 mixing ratio, correct? Ie, NO2 is the limiting reagent and HO2 is in excess.***

The mixing ratio of $HO_2NO_2$ delivered by this source was quantified by TD-CRDS (i.e., the difference in $[NO_2]$ between the heated and room temperature channel.

We added the following sentence on line 194:

"The amount of PNA delivered from this source was quantified by TD-CRDS."

***Line 190-191 – O2 is not readily photolyzed to form O3 by 254 nm – replace with "...generated by photolysis of O2 by 185 nm radiation from a low-pressure mercury lamp"?***

We used a 254 nm lamp as stated in the text - both the 254 and 185 nm version will photo-dissociate O2. The 185 nm one generates way too much $O_3$ for our particular application, however.

We have not amended the manuscript in response to this comment.

***Box model simulations (in SI) The SI discusses formation of C2H5ONO and C2H5O2NO2, but what about the temperature dependence of C2H5ONO2? That is, ethyl nitrate, formed by C2H5O2 + NO.***

The reviewer is referring to section S1.4, where we discuss molecules whose formation is not included in the MCM. Formation of ethyl nitrate is included in the MCM and did not need to be discussed in this section.

For $C_2H_5O_2 + NO \rightarrow C_2H_5ONO_2$, the MCM rate expression is $2.25\times10^{-14}\times e^{(380/T)}$, i.e., the reaction has a negative activation energy and slows down at higher temperatures. The branching ratio (relative to $C_2H_5O_2 + NO \rightarrow C_2H_5O + NO_2$) is 0.9%, which implies that this reaction is an important radical sink especially at higher chain lengths (in the simulations, anyhow).

However, a general problem with using the MCM to simulate TD-PERCA chamber kinetics is that rate constant expressions in the MCM are for atmospheric temperature regimes, i.e., < 300 K. We acknowledged this limitation in section S1.0 "An additional limitations is that the MCM has only been validated at ambient temperature and below, and the rate constants are more uncertain at elevated temperatures."

We agree with the reviewer that the temperature dependence of ethyl nitrate formation may perhaps be worth another look at. However, given that this would be speculation only, we have chosen not to wade into this discussion and have not altered the manuscript in response to this comment.

*Section 3.2 and figure 5. This is overall very good demonstration of the technique. It is a bit confusing that, apparently, both the inlet heater and PERCA chamber can be heated separately. This should be more explicitly pointed out in the earlier experimental sections.*

The reviewer is correct that we can apply heat to two separate inlet sections.

We added the following on line 117:

"The remaining two channels were equipped with heated quartz tubes to monitor $NO_2$ + $\Sigma PAN$ and $NO_2$ + $\Sigma PAN$ + total alkyl nitrates ($\Sigma AN$) (Paul et al., 2009)."

and modified the text in section 3.2 (lines 234-241) as follows:

"A time series demonstrating amplification of PAN and PPN in the TD-PERCA-CRDS  °C is shown in Figure 5. In this experiment, PAN and PPN were delivered via the preparatory-scale GC (Figure 2), and the single-channel setup (section 2.1.1) was used.

PAN and PPN eluted from the GC column after 3 min and 6 min, respectively. The compounds eluted as plateaus because of the relatively long (~30 s) injection time. In Figure 5A, PAN and PPN are observed only by the heated ($NO_2$+ $\Sigma PAN$) TD-CRDS channel. This channel was operated with its quartz inlet at 250 °C  to quantitatively (see Figure 5 of (Paul et al., 2009)) decompose PAN and PPN to $NO_2$ . In this example, mixing ratios of 2.00±0.09 ppbv and 1.86±0.12 ppbv were observed, respectively (errors are 1 σ of 1 s data)."

*Ling 281: the text in the parenthesis, though likely true, makes the sentence awkward to read*

We agree and have removed the text in the parenthesis.

*Section 3.5: interestingly the amplification factor for PNA (yielding HO2) is less than that for PAN (which forms CH3CO3). The following sections address details of the chain length with T, RH, but is there is a conclusion for why the PNA vs. PAN results are so different?*

Yes, the amplification factor itself is temperature dependent. An entire section of text (section 3.5.3) is devoted to this. Specifically, line 350-353 state that "It is obvious from Figure 11 that the amplification factor is strongly dependent on temperature: Even though PNA fully dissociates at temperatures > ~90 °C in our inlets (Figure 4), the amplified signal increases by ~60% in the region from 90 °C to 135 °C (Figure 11, insert), corresponding to amplification factors of ~15 and ~22, respectively. This increase is qualitatively consistent (if extrapolated) with the higher amplification factor observed with PAN or PPN at 250 °C."

We have not amended the manuscript in response to this comment.

*Line 309: ": : :operated under optimal conditions and : : :" – this assumes that the optimal conditions do not change under varying circumstances. Might it be possible that the optimum NO or ethane concentrations are different at different temperatures or RH values?*

It's possible, though we don't believe that the optimum concentrations would change by much. However, since we didn't re-optimize at every RH and temperature, the reviewer has a point that we cannot claim that that TD-CRDS was operated optimally throughout. What we tried to say was that NO and ethane concentrations were constant in these experiments; these mixing ratios are stated in the caption of Figure 9.

We have modified the text on line 312 as follows:

"... the RH dependence was investigated systematically at constant NO and ethane concentrations with TD-PERCA-CRDS operated under optimal conditions and with PAN and PPN at 250 °C inlet temperature. The results are summarized in Figure 9."

*Section 3.7.2: The observed interferences are very interesting, and are likely relevant not only to TD-PERCA-CRDS but also to non-amplified thermal dissociation methods, e.g. TD-LIF.*

This is conceivable, but unlikely. The ethane-PERCA will amplify any molecule that generates trace levels OH, $HO_2$, $RO_2$, RO, or O when heated; in un-amplified TD, concentrations of these radicals are likely too small to have much impact. The only exception is, perhaps, $O_3$, which generates O, that can react with $NO_2$ to NO and $O_2$. Ron Cohen's group is aware of this: In Atmos. Chem. Phys., 14, 12441-12454, 10.5194/acp-14-12441-2014, 2014, Appendix A, Lee et al. discuss the impacts of O3 pyrolysis in their system.

We have not amended the manuscript in response to this comment.

*Section 3.8, discussion of detection limit. Some of the terms here are confusing.*

*1. Do the authors actually mean precision when they have written LOD? LOD needs to be defined – is it for signal to noise ratio of 2? Or 3? The LOD is quoted as 87 ppt (1 sigma, 1 sec), but this seems much more like a description of the precision, not the LOD (ie, 1 sigma for precision, signal to noise ratio for LOD).*

We agree with the reviewer that this section requires a few clarification.

Detection limit is defined by the International Union of Pure and Applied Chemistry (IUPAC) (https://goldbook.iupac.org/html/L/L03540.html) as follows: "The limit of detection, expressed as the concentration, $c_L$, or the quantity, $q_L$, is derived from the smallest measure, $x_L$, that can be detected with reasonable certainty for a given analytical procedure. The value of $x_L$ is given by the equation

$$x_L = \bar{x}_{bi} + k\, s_{bi}$$

where $\bar{x}_{bi}$ is the mean of the blank measures, $s_{bi}$ is the standard deviation of the blank measures, and k is a numerical factor chosen according to the confidence level desired."

In CRDS, $\bar{x}_{bi}$ equals zero, since we subtract the "zero" level from the signal (i.e., $1/\tau_0$ from $1/\tau$). The confidence level for the LOD calculation was stated on lines 25, 445, 446, and 448 as (1 s, 1 $\sigma$); from this, it is straightforward to calculate the LOD by multiplying with the k value of one's choice. However, the reviewer is correct that we should have been more careful here since k = 2 or k = 3 are more commonly chosen. In response to the reviewer's comments, we have the changed the definitions on these lines to (1 s, 2 $\sigma$) and adjusted all values accordingly.

***2. The authors have taken the "LOD" for the CRDS of 87 ppt (1 sigma, 1 s) and divided by the CL of 69 to come up with the LOD for PANs of 1.3 ppt. Realistically, measuring PANs involves measuring NO2 twice in amplification mode and in reference mode (either sequentially in a single channel instrument, or simultaneously with in a multi-channel instrument), so there should probably be another factor of sqrt(2).***

Our apologies as we should have stated how the calculation was made. The statistics we stated are based on time series after subtraction of the reference channel, making multiplication by another factor of √2 unnecessary. We inserted the following on line 448:

"... LOD for ΣPAN* (calculated on the basis of observed precision after subtraction of the reference channel signal, multiplying by √2,  and dividing this precision by the CL) was ..."

We also noted that the precision can vary slightly between days and detection channels and added the following on line 444:

"The precision of the $NO_2$ measurement (and hence the LOD) varied slightly between detection channels and from day to day. Typically, when sampling zero air, the LOD for $NO_2$ was ~100 pptv (1 s, 2 $\sigma$)."

***Also, the authors point out that the precision of the CRDS NO2 measurement is affected by the presence of NO and ethane reagent gases. For measurement in ambient air, or laboratory air, what is the precision of measuring NO2?***

We have only determined the precision under laboratory conditions and stated on line 445 that "In the presence of NO and ethane reagent gases, the LOD was larger, typically ~174 pptv (1 s, 2 $\sigma$)."

In ambient air, $NO_2$ and $O_3$ concentrations vary naturally; the extent of these fluctuations depend on the measurement location. We use parallel detection channels to keep track of (most of) such changes, but we agree that this could still be a source of additional noise. In addition, there may be noise associated with locating the instrument at a field site, where power, temperature, etc. can fluctuate. We have added the following statement on line 452:

"Under field conditions, where NO, $NO_2$ and $O_3$ concentration vary, the LOD is expected to be higher, though this was not evaluated in this work."

*The LOD (and precision) for an actual PNs measurement in ambient air would be affected by the precision of the CRDS NO2 measurement at the actual measurement conditions. For example, if O3 is 25 ppb, some portion of the O3 will react with the NO to give up to 25 ppb NO2 – is the precision the same at 0 ppb and 25 ppb? This has likely been addressed in earlier NO2 CRDS papers but should be mentioned for the reader's sake.*

Please see our response to the preceding comment.

[revised manuscript text omitted]

**S1.1 MCM only simulations**

The chemistry in the PERCA reactor was modelled at temperatures of 25 °C or 250 °C using a subset of the MCM V3.3.1 obtained from http://mcm.leeds.ac.uk/MCM by adding ethane to the marked list and extracting the subset in the Kinetic Preprocessor (KPP) (Sandu and Sander, 2006) format along with inorganic reactions and generic rate coefficients. A plug flow modelling approach was used where initial conditions and mixing ratios were set and the model allowed to proceed without any additional inputs or outputs. Radical wall-loss reactions were not added.

Simulations were performed using the optimized PERCA reagent gas concentrations (i.e., 650 ppbv NO; 1.65 % ethane) at either 25 °C or 250 °C. Table S1 provides an overview of the model inputs. The model runs were also initialized with an initial input of 15, 100, 300, 600, 900, or 1200 pptv of either $HO_2$ or $CH_3O_2$.

**Table S1. Model inputs for simulations run at 25 °C and 250 °C**

| | Mixing Ratio | Number density at 25 °C (molecules cm⁻³) | Number density at 250 °C (molecules cm⁻³) |
|---|---|---|---|
| M | - | $2.14 \times 10^{19}$ | $1.22 \times 10^{19}$ |
| NO | 650 ppbv | $1.39 \times 10^{13}$ | $7.92 \times 10^{12}$ |
| $C_2H_6$ | 1.65% | $3.53 \times 10^{17}$ | $2.01 \times 10^{17}$ |
| $O_2$ | 20.1% | $4.29 \times 10^{18}$ | $2.45 \times 10^{18}$ |
| $N_2$ | 77.6% | $1.66 \times 10^{19}$ | $9.46 \times 10^{18}$ |

[Figure]

**Figure S1. Time series of peroxy radicals (left-hand axis) and hydroxyl radicals (right-hand axis) for a simulation initiated with 15 pptv of $HO_2$ at 25 °C.**

[Figure]

**Figure S2. Time series of peroxy radicals (left-hand axis) and hydroxyl radicals for a simulation initiated with 15 pptv of HO$_2$ at 250 °C.**

Figures S1 and S2 show peroxy and hydroxyl radical concentrations during the first 5 seconds of simulations initiated with 15 pptv of HO$_2$ at 25 °C and 250 °C, respectively. In our chamber, the PERCA reactions are stopped as the radicals encounter the filter after 2.3 s in the dual channel setup. Due to the lower gas density at higher temperatures, a lower concentration of RO$_2$ radicals is present initially (even though the mixing ratio is the same in both cases). Reactions of HO$_2$ and RO$_2$ with NO (e.g., R3, Table 2 of the main manuscript) have negative activation energies and are hence slower at higher temperatures, leading to a lower rate of OH radical production and, since OH loss rates are similar, to lower OH concentrations (maximum of ~$1.9\times10^4$ molecules cm$^{-3}$ at 250 °C vs. $2.2\times10^5$ molecules cm$^{-3}$ at 25 °C) and lower turnover numbers.

Figure S3 shows the CL (number of NO$_2$ molecules produced divided by molecules of RO$_2$ present originally) as a function of temperature and mixing ratio of radicals added initially. Simulations initiated with HO$_2$ radicals gave identical results to simulations initiated with CH$_3$O$_2$ radicals. For simulations conducted at 25 °C, CL is concentration dependent and decrease with increasing concentration. For simulations conducted at 250 °C, the chain length is still concentration dependent, albeit to a much lesser extent than at room temperature. At 250 °C, the CL in the initial 2.3 s of reaction time is well below the CL obtained at 25 °C, inconsistent with experiment which showed a CL of 69±5 at 250 °C (much higher than CL obtained at 25 °C. This suggests that experimental CL are largely a function of wall reactions not included in the simulations.

[Figure]

**Figure S3. Time series of CL for simulations initiated with 100, 300, 600, 900, and 1200 pptv of CH₃O₂ at 25 °C (blue colors) and 250 °C (red colors).**

**S1.2 Inclusion of HO₂.H₂O cluster in the mechanism**

Another factor contributing to observed CL are radical losses due to $HO_2$ water cluster formation (i.e., $HO_2.H_2O$) (Kanno et al., 2006). To estimate the temperature dependent formation of $HO_2.H_2O$ cluster the model assumed that equilibrium between the cluster and $HO_2$ and $H_2O$ was established at every model time step (0.1 s) following the equilibrium rate constant reported by Kanno et al. (2006): $6.6\times10^{-17} \times T \times e^{(3700/T)}$. Upon formation, $HO_2.H_2O$ can react with $HO_2$ ($k = 6.0\times10^{-13}$ 
[revised manuscript text omitted]